# URL: A Representation Learning Benchmark for Transferable Uncertainty Estimates

**Michael Kirchhof**
University of Tübingen
michael.kirchhof@uni-tuebingen.de

**Bálint Mucsányi**
University of Tübingen

**Seong Joon Oh**
University of Tübingen, Tübingen AI Center

**Enkelejda Kasneci**
TUM University

## Abstract

Representation learning has significantly driven the field to develop pretrained models that can act as a valuable starting point when transferring to new datasets. With the rising demand for reliable machine learning and uncertainty quantification, there is a need for pretrained models that not only provide embeddings but also transferable uncertainty estimates. To guide the development of such models, we propose the *Uncertainty-aware Representation Learning* (URL) benchmark. Besides the transferability of the representations, it also meaExamplessures the zero-shot transferability of the uncertainty estimate using a novel metric. We apply URL to evaluate eleven uncertainty quantifiers that are pretrained on ImageNet and transferred to eight downstream datasets. We find that approaches that focus on the uncertainty of the representation itself or estimate the prediction loss directly outperform those that are based on the probabilities of upstream classes. Yet, achieving transferable uncertainty quantification remains an open challenge. Our findings indicate that it is not necessarily in conflict with traditional representation learning goals. Code is available at https://github.com/mkirchhof/url.

## 1 Introduction

Pretrained models are a vital component of many machine learning applications. The driving force behind their development has been representation learning benchmarks, e.g. Roth et al. (2020); Chen et al. (2020): They task models to output representations $e(x)$ of input data $x$ that generalize across datasets in a zero-shot manner. These pretrained representations provide a valuable starting point for downstream applications, requiring less supervised data to be fine-tuned for specific tasks.

At the same time, uncertainty quantification remains a major challenge in the recent efforts towards reliable machine learning (Collier et al., 2023; Tran et al., 2022). Uncertainty quantification refers to estimating the degree of uncertainty or risk $u(x) \in \mathbb{R}$ in a model's prediction. This is particularly important in high-stakes applications such as medical image classification. Here, the model can refrain from making predictions if the uncertainty, e.g., $u(x) := 1 - \max_y P(Y = y|x)$, is too high (Zou et al., 2023; Bouvier et al., 2022). Beyond classification, uncertainty is an inherent property of vision and language (e.g., low image resolution or ambiguous text inputs) that cannot be learned away even with large amounts of data (Chun et al., 2022; Kendall and Gal, 2017). Consequently, recent literature suggests representing images not as points $e(x)$, but as probabilistic embeddings (Kirchhof et al., 2023; Collier et al., 2023; Chun et al., 2021). Here, $u(x)$ is the variance parameter of a distribution around $e(x)$ in the embedding space, representing the input's inherent ambiguity. This can then be utilized for uncertainty-aware retrieval.

37th Conference on Neural Information Processing Systems (NeurIPS 2023) Track on Datasets and Benchmarks.

A major hurdle on the way to reliable uncertainty estimates is that $u(x)$ needs to be trained from the ground up for each specific task, requiring substantial labeled data. Replicating the successes of representation learning promises to reduce this burden by pretraining a $u(x)$ which can be transferred to downstream tasks in a zero- or finetuned few-shot manner. Yet, this transferability of $u(x)$ to new datasets has not been tested in literature, with previous benchmarks evaluating on the same datasets they trained on (Detommaso et al., 2023; Nado et al., 2021). Thus, we propose a novel *Uncertainty-aware Representation Learning* (URL) benchmark. Models that output both embeddings $e(x)$ and uncertainty estimates $u(x)$ of any form are pretrained on large collections of upstream data and evaluated on unseen downstream datasets. The transferability of their embeddings $e(x)$ is evaluated in terms of the Recall@1 (R@1), as in established representation learning benchmarks (Roth et al., 2020; Chen et al., 2020). The transferability of their uncertainty estimates $u(x)$ is evaluated with a novel metric, the Recall@1 AUROC (R-AUROC). It naturally extends R@1-based benchmarks and can be seamlessly integrated in as little as four lines of code, without requiring any new ground-truth labels. Nonetheless, it is not only an abstract metric but has practical significance: Models with higher R-AUROC are also more aligned with human uncertainties and react better to uncertainty-inducing interventions like image cropping.

On this benchmark, we reimplement and train eleven state-of-the-art uncertainty estimators, from class-entropy baselines over probabilistic embeddings to ensembles, with ResNet (He et al., 2016) and ViT (Dosovitskiy et al., 2021) backbones on ImageNet-1k (Deng et al., 2009). Our main findings are:

1. Transferable uncertainty estimation is an unsolved challenge (Section 4.2),

2. MCInfoNCE and direct loss prediction generalize best (Section 4.3),

3. Uncertainty estimation is not always in conflict with embedding estimation (Section 4.4),

4. Models with good uncertainties upstream are not necessarily good downstream (Section 4.5),

5. URL captures how aligned a model is with human uncertainty (Section 4.6).

These findings demonstrate that pretraining models for downstream uncertainty estimation is an important yet unsolved challenge. We hope that our benchmark will serve as a valuable resource in guiding the field towards pretrained models with reliable and transferable uncertainty estimates.

## 2  Related work

Our benchmark connects recent uncertainty quantification benchmarks with representation and zero-shot learning for unseen data, which we introduce below. Specific datasets and methods for uncertainty quantification are described in the experiments section when they are benchmarked.

**Uncertainty benchmarks.**  Uncertainty quantification has become an essential consideration for reliable machine learning, and so several libraries have been recently developed to guide its advancement (Detommaso et al., 2023; Nado et al., 2021). These libraries provide various metrics for evaluating and improving uncertainty estimates on in-distribution data. Galil et al. (2023b) and Galil et al. (2023a) benchmarked over 500 large vision models trained on ImageNet from the timm (Wightman, 2019) library and reported that Vision Transformers (ViT) provide the best uncertainty estimates. Further, scaling of these ViTs to up to 22B parameters and pretraining on a large corpus of upstream data results in very accurate uncertainty estimates (Dehghani et al., 2023; Tran et al., 2022). However, when moving away from in-distribution data, the quality of uncertainty estimates deteriorates (Tran et al., 2022) and we can only expect that they will be generally higher and allow for out-of-distribution detection (Ovadia et al., 2019). This motivates our benchmark: We aim to develop pretrained models that can generalize their uncertainty estimates and discriminate certain from uncertain examples even within unseen datasets. While some works applied their uncertainty estimates to unseen datasets (Cui et al., 2023; Collier et al., 2023; Ardeshir and Azizan, 2022; Karpukhin et al., 2022), their downstream evaluations focused on embeddings, leaving the uncertainty estimates untested. Our benchmark intends to bridge this gap and assess the *transferability of uncertainty estimates*, with the goal of enhancing large pretrained models towards zero-shot uncertainty estimation. To design a benchmark for transferability, we connect the upper benchmarking techniques to paradigms from representation and zero-shot learning below.

```
1   def url_benchmark(pretrained_model, downstream_loader):
2       # Predict embeddings and uncertainties on downstream data
3       labels = embeddings = uncertainties = list()
4       for (image, label) in downstream_loader:
5           embededding, uncertainty = pretrained_model(image)
6           (labels, embeddings, uncertainties).append(label, embedding, uncertainty)
7
8       # Calculate R@1 and R-AUROC
9       next_neighbor_idx = search_most_similar(embeddings)
10      is_same_class = labels == labels[next_neighbor_idx]
11      r_at_1 = mean(is_same_class)
12      r_auroc = compute_auroc(uncertainties, not is_same_class)
13
14      return r_at_1, r_auroc
```

Algorithm 1: Adding URL to existing representation learning benchmarks takes only the four highlighted lines of code.

**Representation and zero-shot learning.** Transferability is generally evaluated by testing whether models can make sensible decisions on unseen data. In zero-shot learning (Xian et al., 2017), the model is tasked to give class-predictions on new downstream classes. This requires both learning a transferable representation space on upstream data and creating classifier heads for the new classes from auxiliary information. Representation learning benchmarks (Roth et al., 2020; Khosla et al., 2020; Bengio et al., 2013) focus on the former. To this end, they use a metric similar to class accuracy, the Recall@1 (R@1) (Mikolov et al., 2013). It calculates the model's embeddings of all unseen downstream data and compares whether each embedding's next neighbor is in the same class or not. This tells whether the embeddings are semantically meaningful, such that the pretrained model can be successfully transferred to downstream tasks. We extend representation learning benchmarks to additionally judge the transferability of uncertainty estimates. To this end, we propose a metric that can be implemented on top of the R@1 in four lines of code in the next section.

## 3 Uncertainty-aware representation learning (URL) benchmark

### 3.1 Evaluating uncertainty about representations

To quantify its uncertainty, a model $f : \mathcal{X} \to \mathcal{E} \times \mathcal{U}$ is assumed to predict both an embedding $e(x) \in \mathcal{E}$ and a scalar uncertainty value $u(x) \in \mathcal{U} \subset \mathbb{R}$ for each input image $x \in \mathcal{X}$. We do not impose restrictions on how $u(x)$ is calculated, e.g., it could be the negative maximum probability of a softmax classifier, a predicted variance from a dedicated uncertainty module, or the disagreement between ensemble members. The predicted uncertainty $u(x)$ is commonly benchmarked in terms of its expected calibration error (ECE), negative log-likelihood, area under the receiver-operator characteristics curve (AUROC), or abstained prediction curves. All of these measures are w.r.t. the correctness of a classification decision. Hence, $u(x)$ can only be evaluated in-distribution with known classes. Our setup involves unseen datasets and classes, so we need to develop a fitting measure.

To this end, let us take Lahlou et al. (2023)'s decision-theoretic perspective on uncertainty quantification: Uncertainty quantification is loss prediction. The uncertainty expresses the expected loss of a model's decision on a specific datapoint. In Gaussian regression with an $L_2$ loss, the expected loss is the target's variance, so an uncertainty quantifier $u(x)$ should be proportional to it. In classification with a 0-1 loss, $u(x)$ should be proportional to the probability of returning the correct class.

In representation learning, the model's decision is the embedding $e(x)$ and the loss is the R@1. The uncertainty quantifier's goal is then to report the loss attached to the embedding, i.e., $u(x)$ should be proportional to whether the R@1 will be correct or not. This demonstrates the use case of $u(x)$: Telling whether an embedding $e(x)$ can be trusted or could be misplaced in the embedding space. This is an important property as models of the form $x \to e \to y$ have an information bottleneck in the quality of the embedding $e(x)$ due to the data-processing inequality (Cover, 1999). For every downstream task, a higher uncertainty $u(x)$ about $e(x)$ monotonically increases the loss of $y(e(x))$. In other words, if the embedding is wrong, then the prediction in any downstream task will be wrong.

We measure whether the uncertainty quantifier $u(x)$ is proportional to the correctness of the embedding $e(x)$ via the AUROC with respect to whether the R@1 is correct (one) or not (zero), named R-AUROC. As the R@1 is a 0-1 loss, the R-AUROC can be interpreted as the probability that an incorrect embedding will receive a higher uncertainty score than a correct embedding (Fawcett, 2006). An R-AUROC close to 1 means that $u(x)$ clearly separates correct from incorrect embeddings, while an R-AUROC of 0.5 means that it has no more predictive power than a random guess.

A positive trait of the R-AUROC is that it is indicative of how well-aligned the model is with human uncertainties and how well the model reacts to uncertainty-inducing interventions (see Fig. 5 and Section 4.6). It also does not require uncertainty ground-truths and takes only four lines of code to implement into existing representation learning benchmarks, as shown in Algorithm 1. We choose the AUROC over other calibration measures such as the ECE because it accepts uncertainties $u(x) \in \mathbb{R}$ (instead of $u(x) \in [0,1]$) and because it avoids some loopholes of the ECE. We discuss these and more design choices behind this metric in Appendix A.

## 3.2 URL benchmark protocol

The R-AUROC can be evaluated on any downstream dataset that allows calculating the R@1, i.e., has class labels. Yet, in order to keep future results comparable, we propose a benchmark protocol for uncertainty-aware representation learning (URL). Our code is based on `timm` (Wightman, 2019) and available at https://github.com/mkirchhof/url.

**Datasets.** We train each model on ImageNet-1k (Deng et al., 2009) as upstream dataset. We note that future works may use larger-scale upstream datasets (Collier et al., 2023; Tran et al., 2022) or auxiliary information (Han et al., 2023; Ortiz-Jimenez et al., 2023), as long as they stay disjoint to the downstream datasets. As downstream datasets, we follow the standardized representation learning protocol of Roth et al. (2020) and use CUB-200-2011 (Wah et al., 2011), CARS196 (Krause et al., 2013), and Stanford Online Products (Song et al., 2016). We follow the original splits that divide their classes into equally sized train and test sets. Following Roth et al. (2020), we further divide the classes in the train set equally into a train and a validation split. In our zero-shot transfering setup, all models are trained only on the upstream ImageNet dataset and do not use the downstream train splits. All results report the performance on the test sets, averaged across the three datasets and three seeds.

**Hyperparameters.** We use the downstream validation split to select the best learning rate, early stopping, and further hyperparameters of each model individually, see also Appendix B. Each model is tuned for 10 search iterations via Bayesian Active Learning (Biewald, 2020). The best model is chosen based on the R-AUROC on validation data, where models with an R@1 below 0.1 on the validation splits are filtered out. The best model is replicated on three seeds.

**Architectures.** Following uncertainty quantification and representation learning benchmarks (Wen et al., 2021; Dusenberry et al., 2020; Roth et al., 2020), we use a ResNet-50 (He et al., 2016) with an embedding space dimension of 2048 as a backbone. We further study ViT-Medium (Dosovitskiy et al., 2021) backbones due to their performance (Galil et al., 2023a,b) and increasing number of large-scale uncertainty quantifiers built on top of them (Collier et al., 2023; Tran et al., 2022). Methods that predict $u(x)$ with explicit modules use a 3-layer MLP head attached to the embeddings.

**Training infrastructure.** Each model is trained with an aggregated batch size of 2048, as recent studies indicate higher batch sizes might benefit uncertainty quantification (Galil et al., 2023b). We use the Lamb optimizer (You et al., 2020) with cosine annealing learning rate scheduling (Loshchilov and Hutter, 2017) for all models since it performed best in preliminary experiments. The ResNets and ViTs are trained on NVIDIA RTX 2080 Ti and A100 GPUs, respectively, for 32 epochs from a checkpoint pretrained on ImageNet to reduce the computational costs. In total, the experiments took 3.2 GPU years of runtime.

**Further metrics.** Uncertainty estimates aim to assess the errors made by individual models, so that they are necessarily model- and performance-dependent. To provide a comprehensive view, we not only evaluate the quality of the uncertainty estimate using R-AUROC but also consider the model's representation learning performance using R@1.

# 4 Experiments

## 4.1 Uncertainty estimators

We apply URL to benchmark two baselines (CE, InfoNCE), five probabilistic embeddings approaches (MCInfoNCE, ELK, nivMF, HIB, HET-XL), two direct variance models (Losspred, SNGP), and two ensembles (Ensemble, MCDropout). We introduce each approach below and explain further details on their reimplementations and hyperparameters in Appendix B and runtimes in Appendix C.7.

**Cross Entropy (CE)** is a supervised baseline which trains under a cross-entropy loss. It uses the entropy of the upstream class probabilities $u(x) := \mathcal{H}(P(Y|x))$ as uncertainty estimate.

**InfoNCE** (Oord et al., 2018) is an unsupervised baseline. Following SIMCLR (Chen et al., 2020), it takes two random transforms of each image and pulls their embeddings towards each other and repels them from the remaining batch. **InfoNCE** itself does not estimate $u(x)$, so we use the embedding norm $u(x) := \|e(x)\|_2$ as a heuristic (Kirchhof et al., 2022; Scott et al., 2021; Li et al., 2021).

**MCInfoNCE** (Kirchhof et al., 2023) follows the unsupervised setup of **InfoNCE**, but predicts a certainty $\kappa(x) =: 1/u(x)$ along with each embedding to define a distribution in the embedding space, so called probabilistic embeddings. It draws samples from them and applies **InfoNCE** on each.

**Expected Likelihood Kernel (ELK)** (Kirchhof et al., 2022; Shi and Jain, 2019) also predicts certainties $\kappa(x) =: 1/u(x)$ to define probabilistic embeddings. The probabilistic embeddings are compared to class distribution via an expected likelihood distribution-to-distribution kernel (Jebara and Kondor, 2003). This makes it a supervised probabilistic embedding-based loss.

**Non-isotropic von Mises-Fisher (nivMF)** (Kirchhof et al., 2022) is analoguous to **ELK**, but models class distributions as non-isotropic von Mises-Fisher distributions, thereby allowing different variances along each embedding space axis. Image certainties are still scalars $\kappa(x) =: 1/u(x)$.

**Hedged Instance Embeddings (HIB)** (Oh et al., 2019) predicts variances $\sigma(x) =: u(x)$ of probabilistic embeddings. Samples are drawn to compute match probabilities between two images. It aims to increase the match probabilities of same-class pairs and decrease that of different-class ones.

**Heteroscedastic Classifier (HET-XL)** (Collier et al., 2023) differs from the above probabilistic embeddings approaches in that it predicts full covariance matrices $\Sigma(x)$ in the embedding space. It draws samples from these probabilistic embeddings to calculate the expected $P(Y|x)$. We test both $u(x) := \det \Sigma(x)$ and the class entropy $u(x) := \mathcal{H}(P(Y|x))$ as possible uncertainty estimates.

**Spectral-normalized Neural Gaussian Processes (SNGP)** (Liu et al., 2020) model class logits as Gaussian Processes with a predicted mean and a heteroscedastic variance. They are pooled into class probabilities $P(Y|x)$ and trained under a CE loss. The entropy of these probabilities serves as uncertainty value $u(x) := \mathcal{H}(P(Y|x))$.

**Loss Prediction (Losspred)** approaches (Upadhyay et al., 2023; Lahlou et al., 2023; Levi et al., 2022; Laves et al., 2020; Yoo and Kweon, 2019) in regression treat uncertainty quantification as secondary regression task. We apply the same principle to classification, where we task the uncertainty module $u(x)$ to predict the (gradient-detached) CE loss at each sample via an $L_2$ loss added to the train loss.

**Deep Ensembles** (Lakshminarayanan et al., 2017) train multiple randomly initiated networks under a CE loss to obtain several predictions. They are pooled one class distribution $P(Y|x)$. We define the uncertainty either via its entropy $u(x) := \mathcal{H}(P(Y|x))$ or the Jensen–Shannon divergence between the ensemble members' class probability distributions. Following (Lee et al., 2015), we only train multiple output heads and share the backbone to reduce computational complexity.

**MCDropout** (Gal and Ghahramani, 2016) applies Dropout (Srivastava et al., 2014) at inference time. This gives multiple predictions per input, imitating the upper Ensemble. We use both the entropy $u(x) := \mathcal{H}(P(Y|x))$ and the Jensen–Shannon divergence between the ensemble members' class probability distributions as possible uncertainty metrics.

## 4.2 Transferable uncertainty estimation is an unsolved challenge

Fig. 1 presents the URL benchmark results, i.e., the R-AUROC calculated for all above approaches on ResNet and ViT backbones. The barplot shows the minimum, average, and maximum performance across three seeds of each hyperparameter-tuned approach.

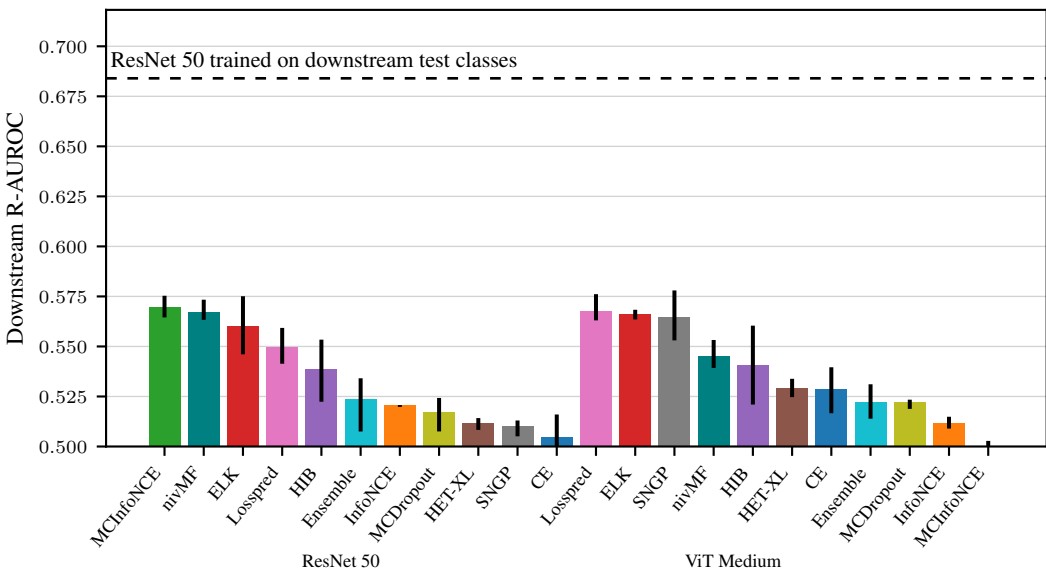

Figure 1: Zero-shot uncertainty estimates of pretrained models (bars) do not reach the performance of many-shot models yet (dashed line). The URL benchmark aims to guide the field to close this gap. Minimum, average, and maximum R-AUROC across three seeds.

Before comparing the models, we first investigate whether the transferability problem URL addresses is already solved by any of the existing methods. To obtain an upper reference, we additionally train a ResNet 50 with standard cross-entropy loss and entropy of the class probabilities as uncertainty prediction on the downstream test classes (split into a train and test split for this experiment only). This many-shot performance of $0.68$ is not reached by any of the methods that transfer their uncertainty in a zero-shot way, marking URL as an open challenge. In the standard R@1, this gap has already been closed in representation learning (see Appendix C.1) and we hope that URL guides the field towards the same for transferable uncertainty estimation.

### 4.3 MCInfoNCE and direct loss prediction generalize best

To compare the approaches in detail, in addition to Fig. 1, Fig. 2 reports both the downstream uncertainty and R@1 performance. The overall best method is Losspred, with the second-best average R-AUROC of $0.568$, close to the best method, **MCInfoNCE**, with an average R-AUROC of $0.569$, while maintaining the second-best average R@1 of $0.53$, close to the best R@1 of $0.57$ achieved by **nivMF**. **MCInfoNCE** marks the best performance in both metrics within the ResNet models, closely followed by **nivMF**. This is remarkable as it is the only unsupervised method aside from the **InfoNCE** baseline. One final noteworthy mention is **ELK** which provides decent uncertainty estimates on both ResNets and ViTs, whereas most other models vary in their performance depending on the backbone.

When grouping the approaches, those that directly model the variance (Losspred, **SNGP**) appear to have an edge on the ViTs, especially Losspred, which is the only method that disentangles variance estimation from how the class logits are calculated. Such disentanglement via having two losses could be added to other approaches in future works. Probabilistic embeddings, especially **MCInfoNCE**, **nivMF** and **ELK**, also show promising performance both on the bigger ViTs and the smaller ResNets. Ensembles fail to provide transferable uncertainty estimates. The baselines unsurprisingly fail, indicating that well-calibrated class probabilities on the upstream dataset do not serve as good uncertainties on downstream data. We investigate this further in Section 4.5.

### 4.4 Uncertainty estimation is not always in conflict with embedding estimation

A commonly raised concern is whether or not uncertainty quantification deteriorates the prediction, or, in the representation learning setup, the embedding quality. In the previous section, we have

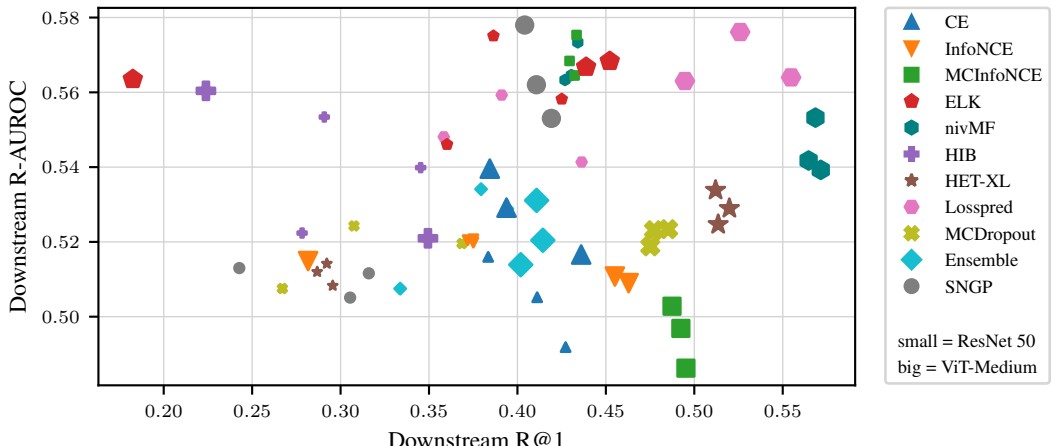

Figure 2: Among ViTs and ResNets, respectively, **Losspred** and **MCInfoNCE** transfer best both in terms of uncertainty estimates (y-axis), measured by our R-AUROC, and embedding quality (x-axis), measured by Recall@1. Three seeds per model and architecture.

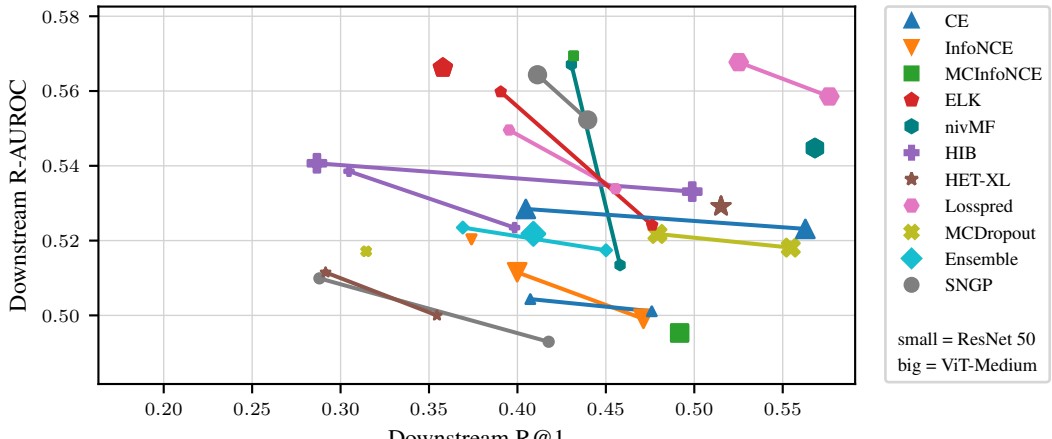

Figure 3: Best hyperparameters chosen for R@1 and for R-AUROC for each model. For some models, there is one best hyperparameter for both, resulting in a point, but most have a large trade-off. Average performance across three seeds.

already seen that **Losspred** can achieve both with only a slight trade-off to the best method in each category. In this section, we further detail this question within each model class.

Fig. 3 shows the performance of the best hyperparameters chosen according to R-AUROC or according to the R@1. If there was no trade-off, the points would lay at the same position or only have a short line connecting them. This is the case for **MCInfoNCE**, **ELK**, **nivMF**, **HET-XL**, and **Ensemble** on ViTs and **MCInfoNCE**, **MCDropout**, and **InfoNCE** on ResNets. The remaining 14 of the 22 approaches show large tradeoffs, e.g., $-0.21$ R@1 for $+0.01$ R-AUROC for **HIB** on ViTs. Whereas this comparison regards only the two extreme ends of the spectrum, Appendix C.2 measures the rank correlation across all tested hyperparameters. It shows a similar conlusion, with 15 out of 22 approaches trading off uncertainty and prediction. However, from another perspective, **Losspred**, **nivMF**, and **MCInfoNCE** are model classes that provide good performance in both simultaneously. Hence, the question of whether there is a general trade-off between uncertainty estimation and prediction is still up to debate. Studying these models and mitigating the model-internal trade-offs is an interesting future work that we hope URL can enable.

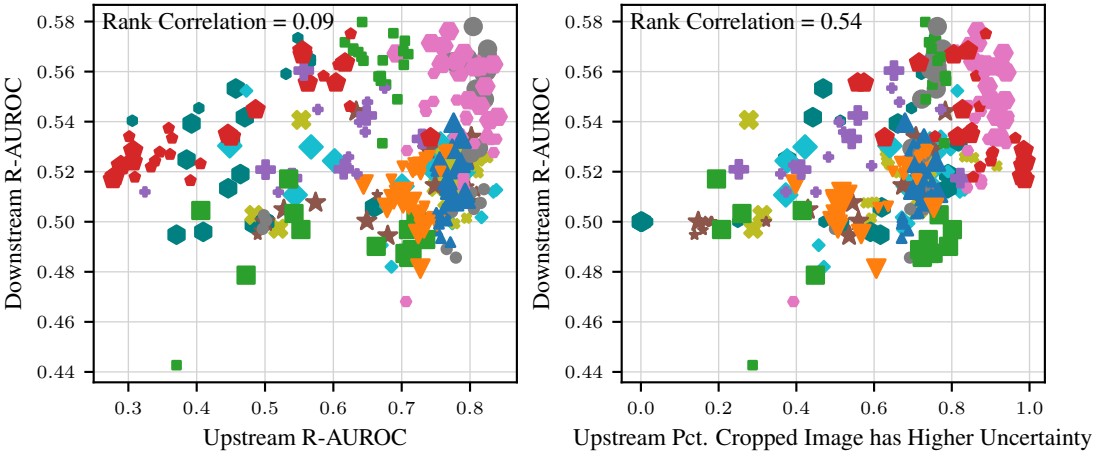

Figure 4: The R-AUROC on upstream data does not indicate the performance on downstream data (left). Yet, the percentage of upstream images where a cropped version receives a higher uncertainty is indicative (right). Plots show all hyperparameters, including non-optimal ones.

### 4.5 Models with good uncertainties upstream are not necessarily good downstream

We have seen that CE is unable to transfer its well-calibrated upstream uncertainty estimates to downstream datasets. This brings up the question of how much the upstream and downstream uncertainty quantification abilities coincide in general. Fig. 4 (left) shows that the majority of models achieves an R-AUROC above 0.7 on the upstream seen classes. But this does not indicate a good downstream performance (Rank Corr. 0.09), neither across nor within model classes (unlike up- and downstream R@1, which transfers better, see Appendix C.3). This demonstrates that transferable uncertainty quantification will not solve itself by merely becoming better on the upstream data.

This also opens a question about model choice: If the upstream performance cannot tell how well the model's uncertainty predictions will perform downstream, how should we select pretrained models? In this paper, we used downstream validation data. However, if we are limited to upstream data, we may test the uncertainty module in a more general task that also holds downstream. In Fig. 4 (right), we calculate how often the model assigns a higher uncertainty value to a cropped version of an image than to the original image. The rank correlation of 0.54 with the downstream R-AUROC signals that models that perform well on this general uncertainty task also tend to generalize better to the downstream data. This means that general uncertainty tasks might be good heuristics to choose models, reinforcing practices in recent literature (Kirchhof et al., 2023).

### 4.6 URL captures how well-aligned a model is with human uncertainty

While the R-AUROC is simple and theoretically founded, readers might still wonder why we want to drive the development of models based on this rather technical-seeming metric. In this section, we show that the R-AUROC reflects how well-aligned the model is with human uncertainties.

To verify this, we use five additional downstream datasets from Schmarje et al. (2022): CIFAR-10H (Peterson et al., 2019), Benthic (Langenkämper et al., 2020; Schoening et al., 2020), Pig (Schmarje et al., 2022), Turkey (Volkmann et al., 2022, 2021), and Treeversity#1 (Arnold Arboretum, 2020). They present human annotators with naturally ambiguous images and record their uncertainty by collecting multiple class annotations per image. The entropy of this distribution measures the human uncertainty $h(x) = \mathcal{H}(P_{\text{human}}(Y|x))$. We can then measure the alignment of the model $f$'s uncertainties with human uncertainties via rank correlation $a(f) = \text{Rank Corr.}(\{u(x), h(x)\}_x)$. Fig. 5 (left) shows that this alignment metric $a(f)$ is positively correlated with the R-AUROC (Rank Corr. 0.80). Further, Fig. 5 (right) shows that the same holds for the correlation between R-AUROC and how well a model detects the uncertainty introduced synthetically via cropping (Rank Corr. 0.71), as in the previous section. This means that the R-AUROC is not just a technical metric, but reveals

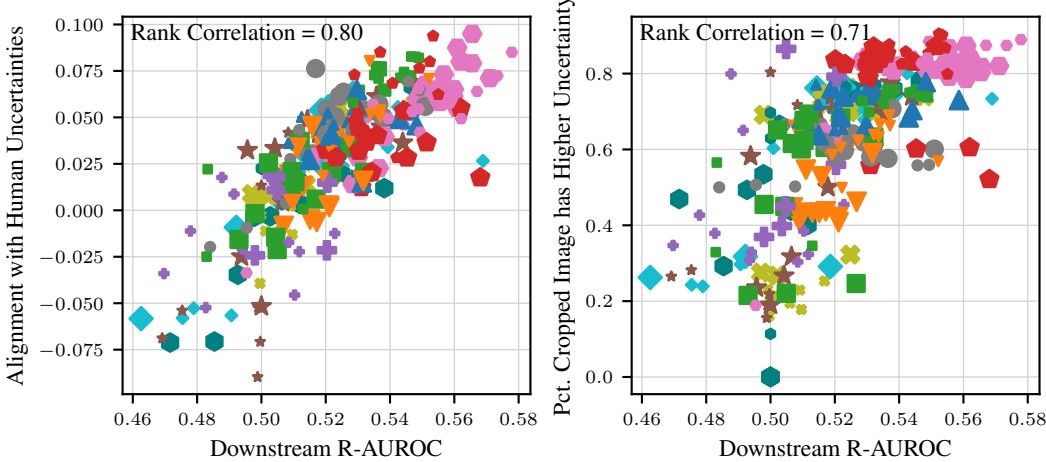

Figure 5: If a model has a high R-AUROC, it is likely also well-aligned with human uncertainties (left). Further, it is likely able to detect uncertainties induced via synthetical cropping (right). All results on five further downstream datasets from Schmarje et al. (2022).

how well a model's uncertainty estimate is aligned with human and synthetical notions of uncertainty, despite not requiring access to human uncertainty ground truths.

### 4.7 URL is no out-of-distribution detection benchmark

Last, we want to clarify how URL is different from out-of-distribution (OOD) detection benchmarks like (Ovadia et al., 2019). While both test uncertainty estimates on OOD data, the goal is different: In OOD detection benchmarks, the uncertainty estimates are tasked to be generally higher for OOD than for in-distribution (ID) samples. In URL, we look only at the OOD data and see if the uncertainties within this data are correctly sorted. This is because our use-case is not to build OOD or anomaly detectors, but pretrained models whose uncertainty estimates generalize to new datasets. In Appendices C.4 and C.5 we show that methods with good OOD detection abilities are not necessarily good in URL or vice versa. This demonstrates that URL is concerned with predictive uncertainty estimation (and generalization), which is largely driven by aleatoric uncertainty, rather than epistemic uncertainty estimation, which is tested in OOD benchmarks.

## 5   Conclusion

**Summary**   This paper proposes the uncertainty-aware representation learning (URL) benchmark. On top of the Recall@1, URL adds an easy-to-implement metric that evaluates how well models estimate uncertainties on unseen downstream data. Besides having a theoretical foundation, it also behaves similarly to practical metrics like the alignment with human uncertainties. In benchmarking eleven state-of-the-art approaches on ResNet and ViT backbones, we found that the challenge URL poses is far from being solved. We hope that URL guides the field to overcome this challenge and yield models with reliable pretrained uncertainty estimates.

**Outlook**   We gathered some insights that might guide future developments: Both unsupervised and supervised methods can learn transferable uncertainty estimates. This is not necessarily at stakes with the embedding and prediction quality. However, many methods have internal trade-offs in their hyperparameters. A deeper analysis of the reasons for this trade-off could allow us to control and mitigate it. Loss prediction and probabilistic embedding methods are currently the most promising approaches. They may be combined to enhance each other and define a new state-of-the-art.

**Limitations**   Although URL allows using any upstream benchmark, we have focused on ImageNet-1k to train all current methods on the same ground. We leave the investigation of further scaled datasets to forthcoming research. Further, we hyperparameter-tuned each model individually with the

same budget, but the vast number of hyperparameters in some models, like SNGP, means that our active learning search may have missed some fruitful combinations. Finally, our study concentrates on zero-shot uncertainty estimates. It will be an interesting endeavour to see if pretrained models with good zero-shot estimates also accelerate learning uncertainties in few-shot scenarios.

## Acknowledgements

This work was funded by the Deutsche Forschungsgemeinschaft (DFG, German Research Foundation) under Germany's Excellence Strategy – EXC number 2064/1 – Project number 390727645. The authors thank the International Max Planck Research School for Intelligent Systems (IMPRS-IS) for supporting Michael Kirchhof.

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

# A    FAQ on URL benchmark and R-AUROC

## A.1    Why AUROC with respect to R@1 and not accuracy?

First attempts in zero-shot learning literature measured the accuracy on unseen classes. This required learning not only an embedder, but also constructing classifier labels, usually through prototypes or attributes. Subsequent approaches in representation and deep metric learning sought generally transferable embeddings, where no information is known about the test classes. So, it was impossible to give class logits and calculate an accuracy. Instead they measure the R@1. We have the same goal of transferring uncertainty estimates to unknown test conditions, so we built up the uncertainty benchmark on predicting whether the R@1 will be correct or wrong. This measures the risk inherent to the embedding, rather than the (classification) downstream task, which is unknown in advance. However, the risk in the embedding is a bottleneck to the potential subsequent classification layer and we indeed observe in Appendix C.3 that R@1 and accuracy are highly correlated, so that the difference is negligible.

## A.2    Why a ranking-based metric (AUROC) and not probability-based calibration?

The AUROC measures if the predicted uncertainties are ranked such that the most uncertain have the most erroneous predictions. It is intriguing to use a different metric, which is to predict the probability of error directly as a value in $[0, 1]$. However, this metric fundamentally cannot be calibrated before the downstream task is known: In classification, we do not know how many classes there will be and beyond classification, our uncertainty metric should indicate the prediction risk, but without knowing the loss or its range in before, we cannot calibrate the uncertainty estimates. Therefore, the best strategy is to give a correctly *ranked* uncertainty metric, which is what AUROC measures. It can be calibrated into the $[0, 1]$ region for the downstream task at hand once data is available. Another point is that a $[0, 1]$ estimate has the trivial solution of always giving a random prediction the certainty being the prior over the classes. A ranking based metric cannot do that, as that would have an AUROC of 0.5. The AUROC thus forces the model to quantify *how* uncertain it is, not only that it *is* uncertain.

## A.3    Does URL measure predictive, aleatoric, or epistemic uncertainty?

The R-AUROC in our setup measures the predictive uncertainty, in other words, the overall expected risk of a prediction. We believe this overall uncertainty is most relevant when seeking a reliable model. Note, however, that the AUROC uses a rank-based relative comparison of the uncertainty estimates. This means that a constant high uncertainty on out-of-distribution data does not suffice; instead, the model needs to quantify how uncertain it is, not just that it is uncertain. This is influenced both by the inherent ambiguity of the downstream sample (aleatoric) and how far it is from the upstream data (epistemic).

## A.4    How is the AUROC implemented precisely?

We use the `TorchMetrics` implementation (Detlefsen et al., 2022). It applies the trapezoidal rule and uses every uncertainty value as a possible threshold.

## A.5    Why CUB200-2011, CARS196, and Stanford Online Products?

Our definition of URL is agnostic of the datasets and can be applied to any downstream dataset. For this paper, we use CUB200-2011 (Wah et al., 2011), CARS196 (Krause et al., 2013), and Stanford Online Products (Song et al., 2016), because they are commonly used as zero-shot learning and representation learning datasets. We hope that this status prevents data-leakage to upstream pretraining datasets.

## A.6    Can URL also be implemented for downstream datasets that don't have labels?

Throughout this paper, we measure the R@1 by seeing if the next neighbor of a test image has the same class label. This is to ensure the compatibility with representation learning benchmarks. One can also define a self-supervised R@1 (e.g., seeing if a crop of the same image is detected as belonging

to that image). In this case, R-AUROC automatically generalizes to this form of supervision, as it still measures errors in R@1, regardless of how it was computed.

### A.7 Can URL be applied beyond vision tasks?

Yes. URL can be deployed whenever there is a downstream dataset on which we measure a R@1.

## B Reimplementations and hyperparameters

All methods benchmarked in this paper are re-implemented and hyperparameter tuned to ensure a consistent comparison. First, let us explain the hyperparameters shared by all approaches: As backbones, we use ResNet-50 (He et al., 2016) architectures with 224x224 inputs and 2048-dim max-pooled embeddings, and ViT-Medium (Dosovitskiy et al., 2021) with an input size of 256x256 split into 16x16 patches. All models are tuned for 32 epochs with a Lamb optimizer (You et al., 2020). 16 minibatches of size 128 are accumulated to reach a summed batch size of 2048, unless otherwise noted. The learning rate is fixed at the first epoch, then warmed up for five epochs and cooled down using a cosine scheduler (Loshchilov and Hutter, 2017). The learning rate is a hyperparameter for all approaches, searched over a range of $[0.0001, 0.01]$. The hyperparameter search is extended by more runs if the optimal value is close to a boundary. Let us now describe the approaches and their additional hyperparameters in detail. All optimal hyperparameters are reported in the code appendix.

### B.1 Cross-Entropy (CE)

The **CE** baseline applies a cross-entropy loss to the class-logits output by a final linear layer appended to the embeddings. It has no hyperparameters except the learning rate.

### B.2 InfoNCE

**InfoNCE** (Oord et al., 2018) uses two random crops per input image that are considered positive, whereas the remaining batch is considered negative. This results in a doubled VRAM usage, so that the batch size is reduced to 21 minibatches of 96 images each. **InfoNCE** uses an (inverse) temperature parameter $t$, tuned within $[8, 64]$. All embeddings $e$ are normalized to lay on the unit sphere, where $e_1^+$ and $e_2^+$ denote the positive and $e^-$ all negative embeddings. The final loss is

$$\mathcal{L} = -t \cdot e_1^{+\top} e_2^+ + \log \sum_{e^- \in \text{Batch}} \exp\left(t \cdot e_1^{+\top} e^-\right) + \log \sum_{e^- \in \text{Batch}} \exp\left(t \cdot e_2^{+\top} e^-\right) . \quad (1)$$

### B.3 MCInfoNCE

**MCInfoNCE** (Kirchhof et al., 2023) works similar to **InfoNCE**, except that it does not directly compare the embeddings, but samples from estimated posteriors $s_{1,i}^+ \sim \text{vMF}(e_1^+, \kappa(e_1^+))$, $s_{2,i}^+ \sim \text{vMF}(e_2^+, \kappa(e_2^+))$, $s_i^- \sim \text{vMF}(e^-, \kappa(e^-))$. The concentration, i.e., inverse uncertainty, $\kappa$ is estimated from a 3-layer MLP attached to the embeddings. Its initial value is either randomly initialized or warmed up to $0.001$, which is a hyperparameter. The second hyperparameter is its (inverse) temperature $t \in [8, 64]$. Like **InfoNCE**, it uses a reduced batch size of 21 times 96. The loss is obtained by calculating the **InfoNCE** loss for 16 samples $s_i$ from the respective posteriors.

$$\mathcal{L} = \frac{1}{16} \sum_{i=1}^{16} -t \cdot s_{1,i}^{+\top} s_{2,i}^+ + \log \sum_{s_i^- \in \text{Batch}} \exp\left(t \cdot s_{1,i}^{+\top} s_i^-\right) + \log \sum_{s_i^- \in \text{Batch}} \exp\left(t \cdot s_{2,i}^{+\top} s_i^-\right) \quad (2)$$

### B.4 Expected Likelihood Kernel (ELK)

**ELK** uses a ProxyNCA formulation, as proposed in Kirchhof et al. (2022). Like **MCInfoNCE**, it parametrizes posteriors $\zeta = \text{vMF}(e, \kappa(e))$ from each image's normalized embeddings $e$ and a 3-layer MLP for $\kappa$. Its initial value is either randomly initialized or warmed up to $0.001$, which is a hyperparameter. **ELK** is supervised and learns vMF class distributions $\rho_c$ for each class $c = 1, \ldots, C$. The concentrations of these classes are scaled up by the hyperparameter $t \in [8, 64]$, which takes a

similar role to the inverse temperature in the previous losses. These are compared to the embedding posteriors via a distribution-to-distribution similarity function `elk_sim`. This is solved analytically, not requiring sampling. With $\rho^*$ denoting the true class of the given sample, the loss can be written as

$$\mathcal{L} = -\texttt{elk\_sim}(\zeta, \rho^*) + \log \sum_{c=1}^{C} \exp\left(\texttt{elk\_sim}(\zeta, \rho_c)\right) . \tag{3}$$

### B.5 Non-isotropic von Mises-Fisher (nivMF)

nivMF (Kirchhof et al., 2022) has the same hyperparameters and loss as ELK, except that the class proxy distributions $\rho_c$ are non-isotropic vMFs. Since the expected-likelihoood between the image embeddings' vMFs and the classes' non-isotropic vMFs has no analytical solution, it is Monte-Carlo approximated with 16 samples.

$$\mathcal{L} = -\texttt{approx\_elk\_sim}(\zeta, \rho^*) + \log \sum_{c=1}^{C} \exp\left(\texttt{approx\_elk\_sim}(\zeta, \rho_c)\right) . \tag{4}$$

### B.6 Hedged Instance Embeddings (HIB)

HIB (Oh et al., 2019), like MCInfoNCE, takes samples from estimated posteriors $s_{n,i} \sim$ vMF$(e_n, \kappa(e_n))$, where $e_n$ are the image's $L_2$ normalized embeddings $e_n$ and $\kappa$ is estimated by a 3-layer MLP. Its initial value is either randomly initialized or warmed up to 0.001, which is a hyperparameter. HIB then calculates a matching probability by comparing the samples of every pair of images $n, m$ in the batch: $p_{n,m} = \sum_{i=1}^{16} \text{sigmoid}\left(t \cdot s_{n,i}^{\top} s_{m,i} + b\right)$, where $t \in [8, 64]$ is a hyperparameter similar to the (inverse) temperature and $b \in [-8, 8]$ is a second hyperparameter. The matching probability should be high for images with the same label and low for images with different labels. Let $\mathcal{I}_{\text{same}}$ denote the pairs of images with the same label and $\mathcal{I}_{\text{different}}$ the pairs with different labels, both without self-matches. Then the loss is

$$\mathcal{L} = -\frac{1}{|\mathcal{I}_{\text{same}}|} \sum_{(n,m) \in \mathcal{I}_{\text{same}}} \log p_{n,m} + \frac{1}{|\mathcal{I}_{\text{different}}|} \sum_{(n,m) \in \mathcal{I}_{\text{different}}} \log p_{n,m} , \tag{5}$$

where $|\cdot|$ denotes the cardinality of the set. As opposed to the original implementation, we use cosine distances instead of $L_2$ distances and remove the prior regularizer. This is to make HIB more comparable to the other approaches in this paper. We also changed the second term from encouraging low log match probabilities for different labels $(-\log(1 - p_{n,m}))$ to discouraging high ones $(+\log p_{n,m})$, which stabilized training. HIB requires additional VRAM and thus uses 21 batches of size 96 (43 of size 48 on ViTs).

### B.7 Heteroscedastic Classifier (HET-XL)

HET-XL (Collier et al., 2023) predicts a distribution in the embedding space for each image. It then takes samples and calculates a Monte Carlo estimate of the expected model output under the embedding distribution. As opposed to the other probabilistic embedding approaches from above, it operates in Euclidean space by predicting a Gaussian distribution $\mathcal{N}(\phi(x; \theta), \Sigma'(x; \theta_{\text{cov}}))$ with a low-rank approximation of the covariance matrix $\Sigma'(x; \theta_{\text{cov}}) = V(x)^{\top} V(x) + \text{diag}(d(x))$. $V(x)$ and $d(x)$ are output by a linear layer attached to the embeddings. The number of columns in $V$ increases the rank of the low-rank approximation, but also the number of parameters that the final linear layer has to predict, and thus the memory requirements. We thus set this hyperparameter to 1 (exploratory experiments with a rank of 10 did not show increased performance). The final loss is

$$\mathcal{L}_{\text{cross-entropy}}\left(\mathbb{E}_{\epsilon'}\left[\text{softmax}_{\tau}(W^{\top}(\phi(x; \theta) + \epsilon'(x)))\right], y\right) \quad \text{with} \quad \epsilon'(x) \sim \mathcal{N}\left(0, \Sigma'(x; \theta_{\text{cov}})\right) , \tag{6}$$

where the softmax temperature $\tau$ is a learnable parameter.

### B.8 Spectral-normalized Neural Gaussian Processes (SNGP)

SNGP (Liu et al., 2020) predicts a Gaussian distribution

$$\mathcal{N}\left(\phi(x)^{\top} \beta, \phi(x)^{\top} \left(I + \Phi^{\top} \Phi\right)^{-1} \phi(x)\right) \tag{7}$$

over the class logits, which is cast into class probabilities via a mean-field approximation. These class probabilities are then trained under a **CE** loss. $\beta$ is a learnable parameter matrix, $\phi(x) = \cos(Wh(x) + b)$ is a feature embedding based on frozen random parameters $W$ and $b$, and $\Phi^\top \Phi$ is the empirical covariance matrix of the feature embeddings over the training dataset. The method also applies spectral normalization to the hidden weights in each layer in order to satisfy input distance awareness. We treat whether to apply spectral normalization through the network and whether to use layer normalization in the last layer as hyperparameters.

### B.9 Direct Loss Prediction (Losspred)

**Losspred** trains a classifier under a cross-entropy loss and uses its uncertainty module $\kappa$, a 3-layer MLP attached to the embedding space, to predict the value of the cross-entropy loss. Both components are balanced by the hyperparameter $\lambda \in [0.01, 0.99]$:

$$\mathcal{L} = \mathcal{L}_{\text{cross-entropy}}(x, y) + \lambda \left( \kappa(x) - \mathcal{L}_{\text{cross-entropy}}^{\text{detached}}(x, y) \right)^2 . \tag{8}$$

The gradients of $\mathcal{L}_{\text{cross-entropy}}^{\text{detached}}$ inside the $L_2$ loss are detached to prevent fitting it to $\kappa(x)$, instead of the other way around. Besides $\lambda$, a second hyperparameter is whether or not to warm up $\kappa$ to $0.001$.

### B.10 Deep Ensemble

**Ensemble** (Lakshminarayanan et al., 2017) has 10 classifier heads attached to the embedding space. Their logits are transformed into probabilities by a softmax and then averaged. This average is trained under a **CE** loss. **Ensemble** has no hyperparameters other than the learning rate.

### B.11 MCDropout

**MCDropout** (Srivastava et al., 2014) trains with a dropout rate of $[0.05, 0.25]$, which is a hyperparameter. During inference time, it keeps the dropout activated to sample 10 logits and averages them like **Ensemble**.

## C Additional results

### C.1 Pretrained models already close the gap in terms of R@1

This section compares the models in terms of their R@1. To this end, we hyperparameter-tuned them with respect to R@1, and not R-AUROC. Similar to the main text, we also add an additional baseline that was trained on the downstream classes, where the original test split was split into equal sized train and test splits.

Fig. 6 shows these performances. As opposed to the R-AUROC, we can see that the gap between pretrained zero-shot and many-shot models is much tighter in terms of the R@1. The best pretrained ResNet-50 has an average R@1 of $0.48$ vs. $0.54$ when training on the downstream data.

Surprisingly, when it comes to R@1, **CE** is among the best two approaches both for ResNet and ViT backbones. In fact, three of the four best approaches on ViTs and two of the four on ResNets rely on **CE** as part of their loss. The approaches that do not rely on **CE** are probabilistic embeddings – **nivMF** on ResNet and ViT and **ELK** on ResNet.

### C.2 R@1 and R-AUROC correlate negatively in most models

In extension to the plot that compared the best hyperparameter setup for R-AUROC to the best for R@1, Fig. 7 and Fig. 8 present the trade-offs for all tested hyperparameters.

The general picture is the same as in the comparison of best vs best: Most models show a trade-off between achieving the best R-AUROC and the best R@1. This is indicated by a negative rank correlation in 15 out of 22 models. Still, there are some models where the uncertainty estimation and prediction performances correlate moderately positively ($0.4 \leq$ Rank Corr. $\leq 0.72$). These are **InfoNCE** and **MCInfoNCE** on ResNets, and **HET-XL**, **nivMF**, and **HIB** on ViTs.

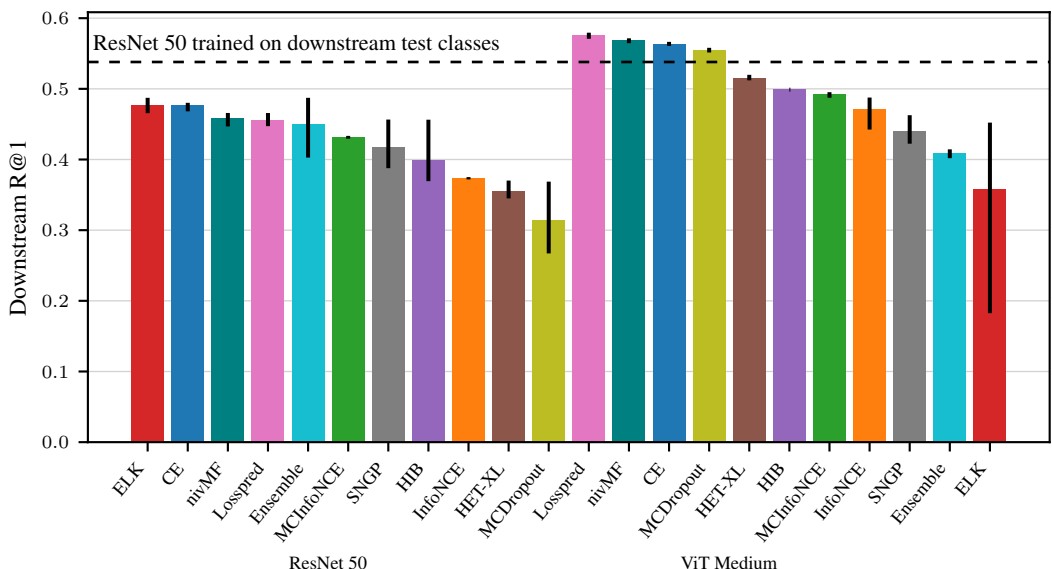

Figure 6: Pretrained models achieve an almost as good R@1 on unseen downstream data as models trained on the downstream data. Minimum, average, and maximum R@1 per model. Model hyperparameters were optimized w.r.t. R@1.

As mentioned in Appendix B, each approach was tested on 10 hyperparameters, with 2 additional runs on other seeds for the best hyperparameters. The reason that some plots show less than 12 points is that those had a R@1 < 0.1 on the downstream datasets and were excluded from the analysis. Some plots also show more than 12 points. This is because their best hyperparameter for R@1 was unlike that for R-AUROC, adding another 2 runs on different seeds for the R@1. Further, some approaches had optimal hyperparameters close to the original search bounds, such that the search was extended, leading to additional points in the plots.

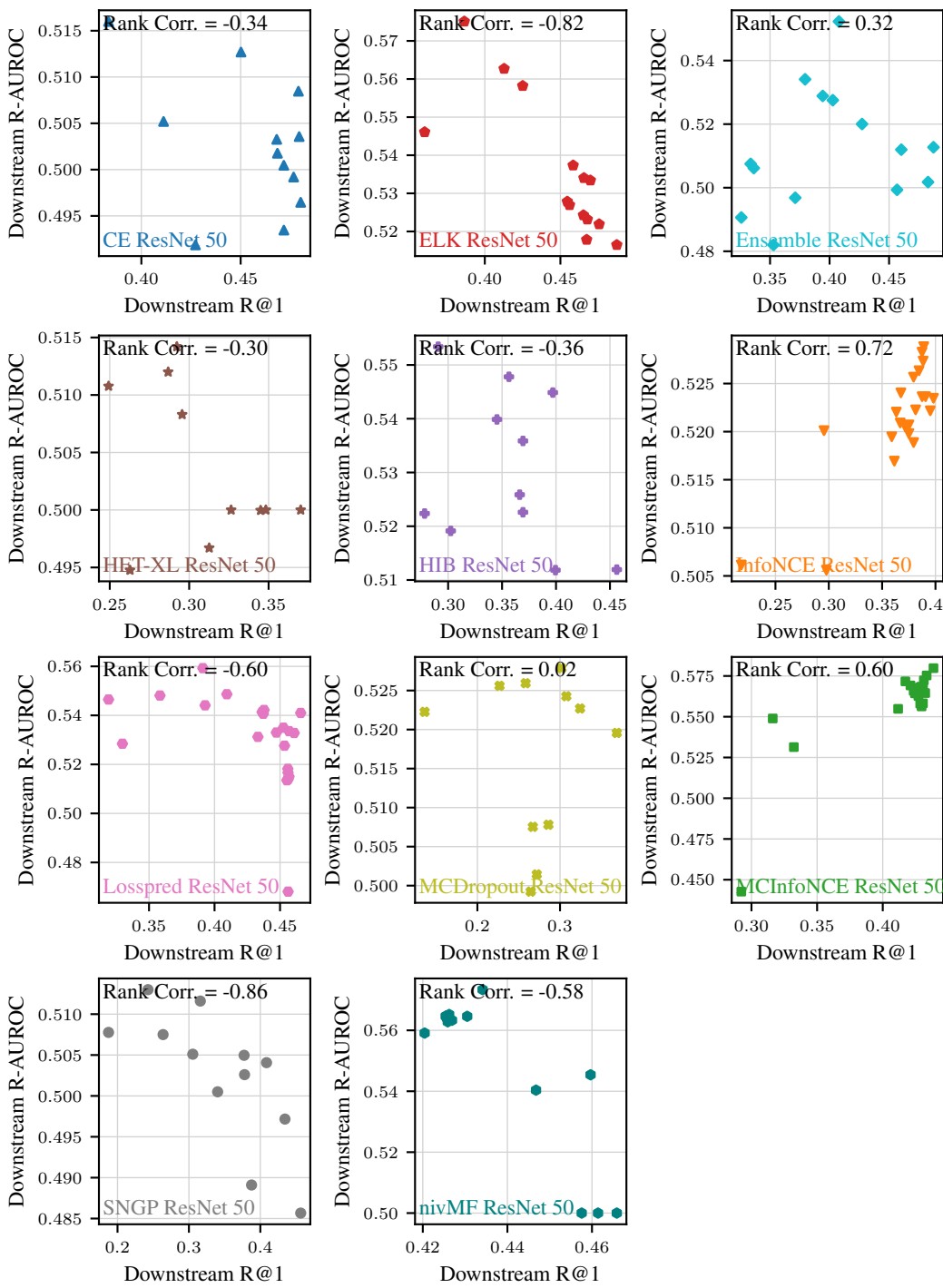

Figure 7: R@1 and R-AUROC are negatively correlated on seven out of ten model classes with ResNet backbones. Plot shows all tested hyperparameter combinations.

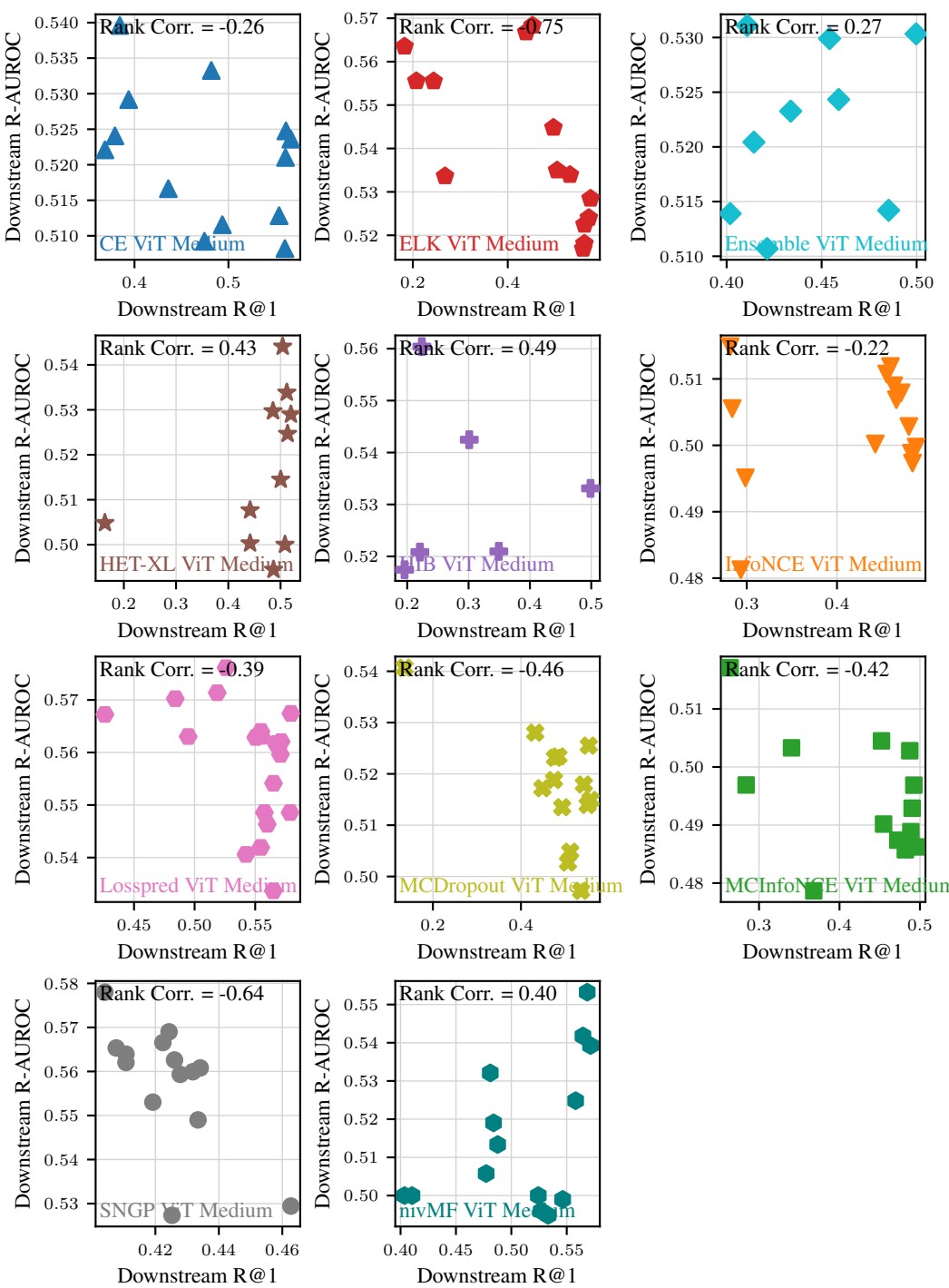

Figure 8: R@1 and R-AUROC are negatively correlated on seven out of ten model classes with ViT backbones. Plot shows all tested hyperparameter combinations.

## C.3 Further correlation of up- and downstream metrics

This section reports further metrics for each model. These include a new metric (top-1 accuracy on upstream ImageNet) and all downstream metrics (R@1, R-AUROC, percentage where cropped image has higher uncertainty) on the upstream dataset.

Fig. 9 shows the pairwise correlations between all metrics, for every tested hyperparameter setting. Let us first consider the new metric, ImageNet top-1 accuracy, on its own. The best-performing models are a **CE** and a **nivMF** ViT, with an accuracy of $0.84$ each. Other than them, **Losspred** and **HET-XL** on ViTs also have a high accuracy, similar to the results on the R@1 benchmark.

Regarding correlations between metrics, accuracy and R@1 are highly correlated, as expected. Further, models with a high accuracy or R@1 on ImageNet also have a high R@1 on the downstream data, resulting in the small gap explained in Appendix C.1. While up- and downstream R-AUROC do not correlate, up- and downstream percentage of cropped images having a higher uncertainty correlate nearly linearly. This reinforces that the percentage can be considered as a general notion of uncertainty. It should, however, be noted that **ELK** ViTs already achieve a performance of $0.99$ on this metric, even on downstream data, such that it is not able to guide the field as well as R-AUROC.

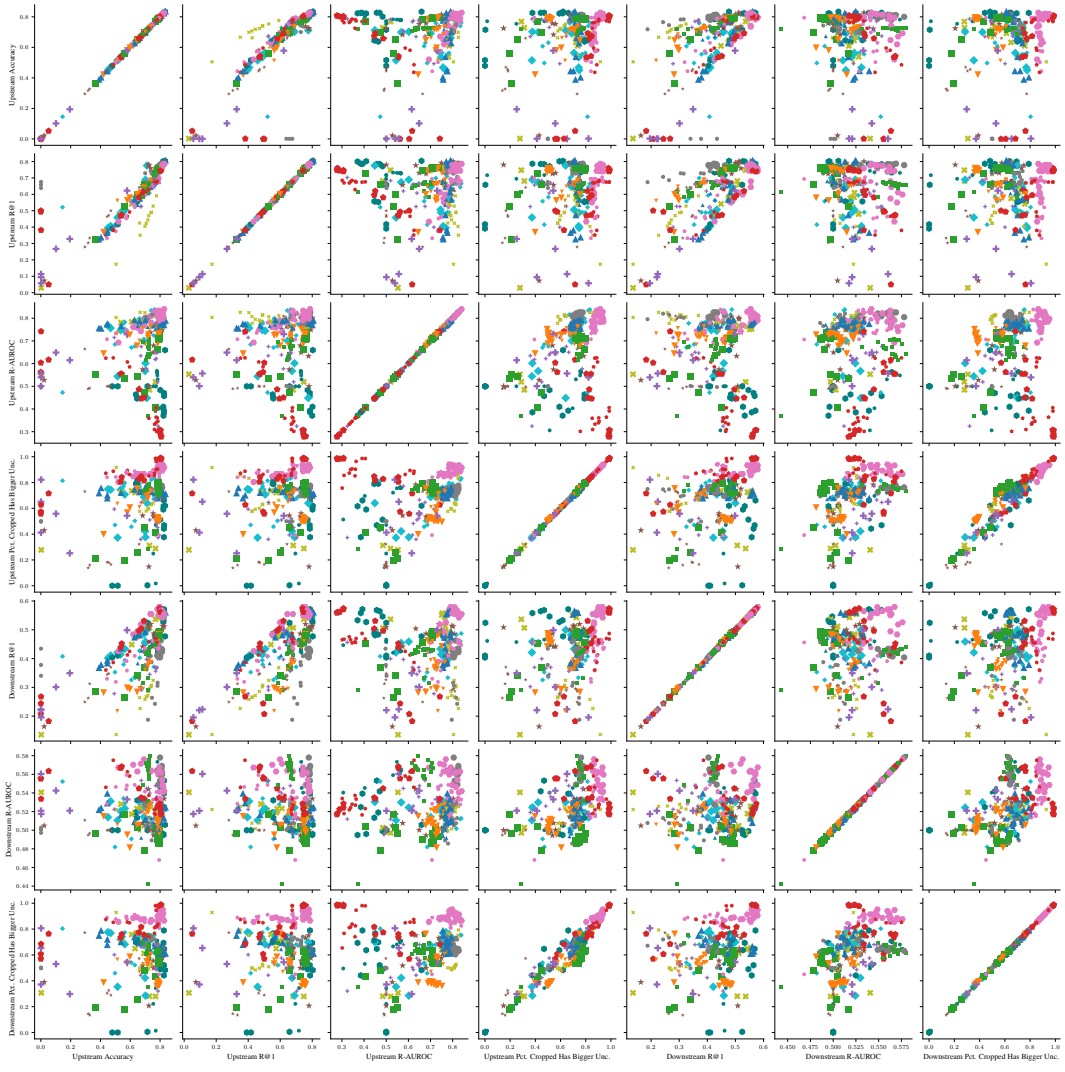

Figure 9: Correlations between several up- and downstream metrics across all models and hyperparameter choices.

## C.4 Class-entropy is useful for OOD detection

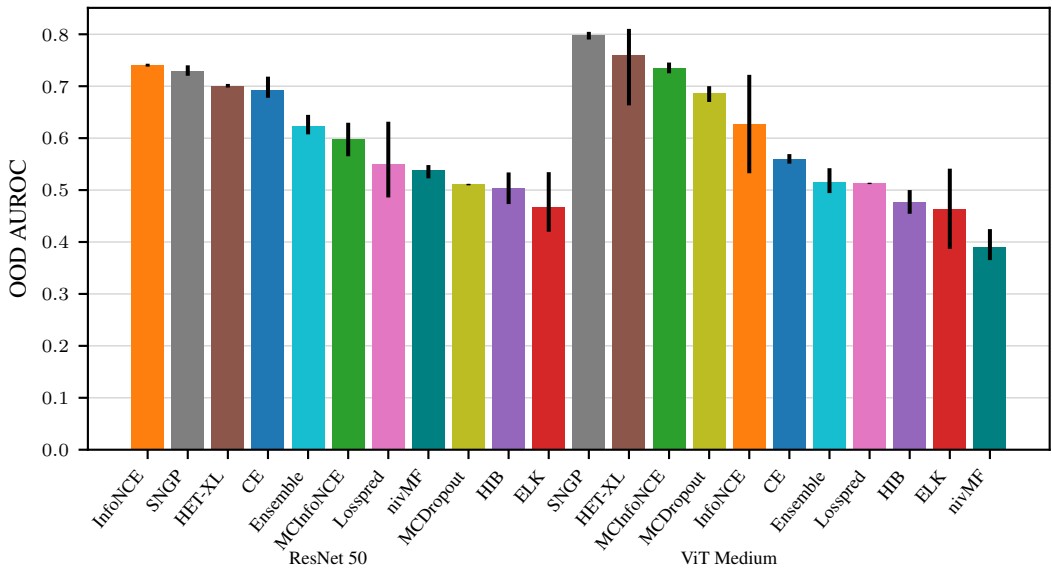

Figure 10: Cross-entropy and embedding norm-based uncertainty estimators have the best OOD detection capabilities. Averaged AUROC of distinguishing ImageNet vs CUB, ImageNet vs CARS, and ImageNet vs SOP. Error bars indicate minimum/maximum performance across seeds.

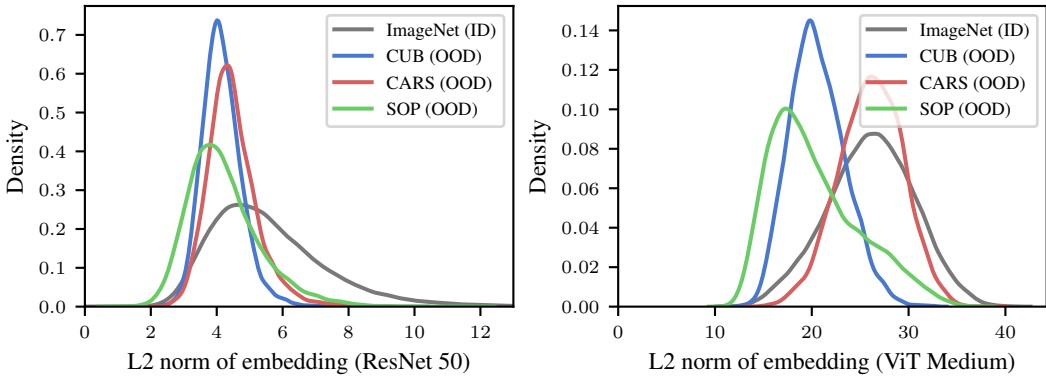

Figure 11: Embeddings of OOD images tend to have smaller $L_2$-norms than embeddings of ID images. Embedding norms are from InfoNCE models that use the (inverse) norm as uncertainty estimate (without using them during training) and reach an OOD AUROC of 0.73.

In this section, we study if uncertainties on downstream data are generally higher than for the upstream data. To this end, we perform out-of-distribution detection experiments: Using the same pretrained models as before, we calculate their uncertainties on downstream dataset samples and on an equally sized set of upstream samples. We quantify how well the predicted uncertainties distinguish whether the sample was from a downstream dataset (1) or not (0) by calculating the AUROC on ImageNet vs CUB, ImageNet vs CARS, and ImageNet vs SOP, and averaging them across the datasets.

Fig. 10 shows the average result across all seeds, along with minimum and maximum performances. Generally, pretrained models perform differently than in the URL benchmark in the main text. This is because the OOD task benchmarks epistemic uncertainty (i.e., OOD data just has to have generally high uncertainties). On the other hand, the URL benchmark tests predictive uncertainties on OOD data. There, it is not enough to predict high values on OOD data but the models need to differentiate within them, which lays more focus on aleatoric uncertainty. In more detail, models that directly predict variances or losses do not provide as good OOD performance. Models that use class-entropy

as uncertainty estimates (**SNGP**, **HET-XL**, **CE**, **Ensemble**, and, only on ViTs, **MCDropout**) and also **InfoNCE**, which uses the norm of the embedding vector, do work well.

There is an intuitive explanation for this. The latter models implicitly provide epistemic uncertainty estimates by construction: ResNets and ViTs embed inputs with less known features closer to the origin. As an example, Fig. 11 shows that for **InfoNCE**, the embedding norms of OOD samples are smaller. Note that this is no trained behaviour; **InfoNCE** only trains on normalized embeddings, so the norms occur naturally. In **InfoNCE**, we use this embedding norm directly as uncertainty estimator. But the same happens in models that use the class entropy: Embeddings with small norm lay close to the origin, where they lead to uniform distributions over the classes, i.e., a high entropy.

In summary, the OOD experiment reveals that probabilistic embeddings and loss prediction methods provide aleatoric uncertainty estimates, whereas models that explicitly or implicitly use the embedding norm provide good epistemic uncertainty estimates. Some models, like **SNGP**, provide a mixture and are good in both tasks.

### C.5 Uncertainties on mixtures of in- and out-of-distribution data

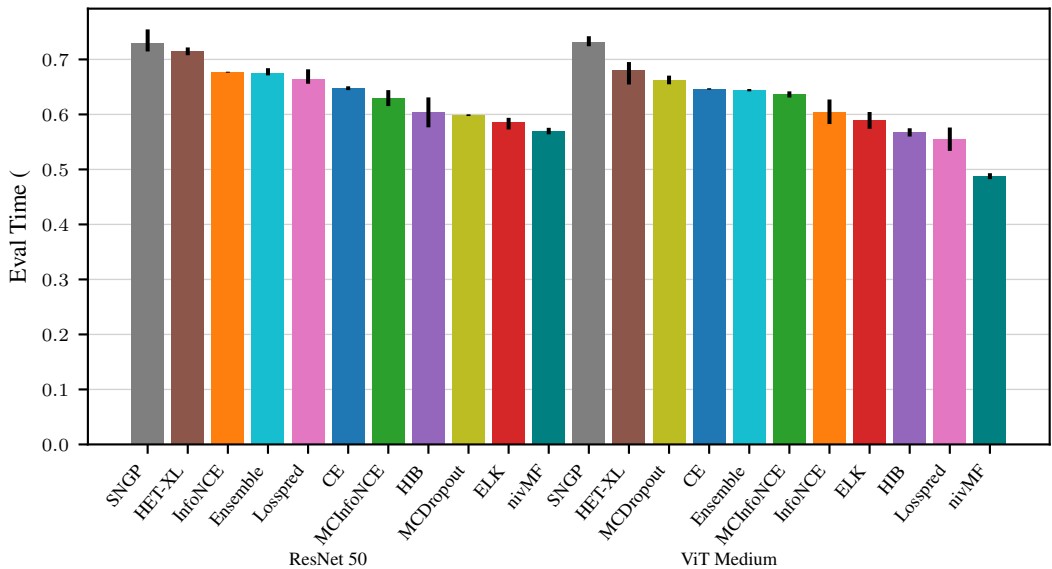

Figure 12: Cross-entropy based uncertainties provide the best mix of OOD detection capability and ordering within ID and OOD uncertainties. R-AUROC on mixed ImageNet+CUB, ImageNet+CARS, and ImageNet+SOP data. Error bars indicate minimum/maximum performance across seeds.

In some applications, models might encounter a mixture of in-distribution and out-of-distribution data. In this case, the model both needs to assign higher uncertainties to the OOD data and it needs to differentiate the uncertainties within both the ID and the OOD sets. This blends the OOD detection of the previous experiment with traditional ID calibration and the OOD calibration of the URL benchmark.

Fig. 12 measures the R-AUROC on a mixture of upstream and downstream data. As in the previous experiment, we use 50/50 splits of ImageNet+CUB, ImageNet+CARS, and ImageNet+SOP, and average across those combinations. We find that **HET-XL** and **SNGP**, which previously performed well on both OOD detection and the URL benchmark, also perform well on this mixed task. The remaining models tend to follow the ranking of the OOD benchmark rather than that of URL. This indicates that good epistemic uncertainty estimation outweighs aleatoric uncertainty estimation in this task. This is intuitive, because the R-AUROC measures how likely it is that a wrong prediction has a higher uncertainty than a correct one. In such a 50/50 split of ID and OOD data, the capability to distinguish ID from OOD data, and thus data with generally less errors from data with generally more errors, thus leads to a higher R-AUROC.

This serves to demonstrate that the quality of an uncertainty estimate always depends on the task and setup at hand. While OOD detection and ID calibration are ideally both reflected within the very same predictive uncertainty value, both are of different importance depending on the data mixture.

## C.6 Few-shot uncertainties starting from pretrained models

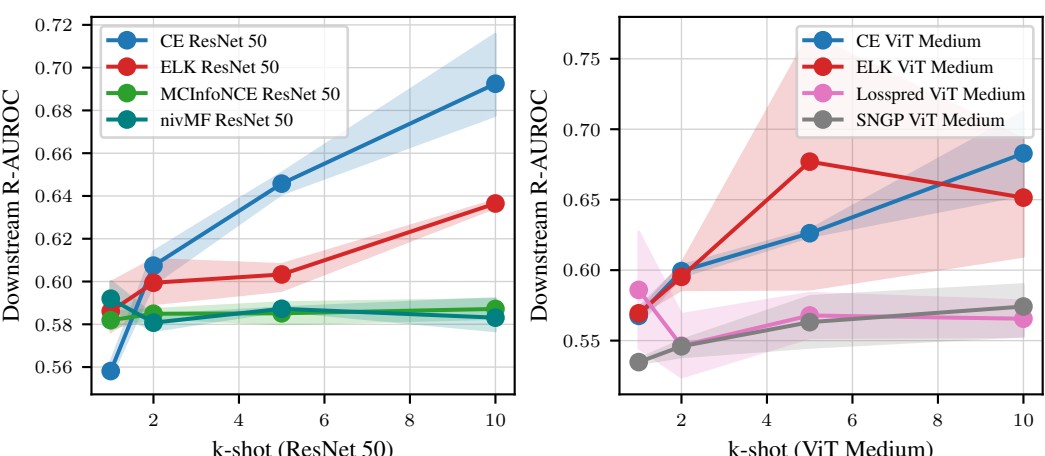

Figure 13: Cross-entropy training on the downstream test data reaches the zero-shot AUROC of 0.6 at 2-5 samples. R-AUROC of pretrained models when finetuned on the downstream test classes.

In this section, we provide an initial attempt on a few-shot experiment. We use both the best three pretrained models (**ELK**, **MCInfoNCE**, and **nivMF** for ResNet 50; **ELK**, **Losspred**, and **SNGP** for ViT Medium) along with the **CE** model as pretrained checkpoints and continue to train under the same hyperparameters and losses on $k \in \{1, 2, 5, 10\}$ samples of each class of the downstream datasets. This is done on the test classes of the downstream classes, which were separated into disjoint train and test samples for this experiment. We train on each CUB/CARS/SOP separately and average their performance.

Fig. 13 shows the minimum, maximum and average performance across the seeds. First, we can see that the normal cross entropy training reaches the zero-shot R-AUROC of the currently best pretrained models at around 2 samples per class (totaling in 200 samples on CUB, 196 on CARS, and 22636 on SOP). This again demonstrates the point that the challenge URL addresses is unsolved yet. It also shows that knowledge of the specific downstream task can increase the uncertainty estimators quality even at only a few samples. Second, we find that most pretrained models increase their performance as well. This, however, happens not for all models and is highly noisy. We attribute this to the simple setup of our experiment, which uses the same hyperparameters as for the pretraining and performs standard training as opposed to specialized few-shot methods. Finding best practices to tune pretrained uncertainties to downstream few-shot tasks is a promising undertaking for future research.

## C.7 Uncertainty estimation does not add significant computational costs

Table 1 reports the computational complexity during all benchmarks. It shows the number of parameters as proxy for RAM usage, the duration of the first training epoch and evaluation, and the time needed for each sample during evaluation. These results were collected on the go and there are possible confounders such as network storage workload. We thus recommend to interpret them as rough indicators. Note also that ViTs were run on NVIDIA A100 GPUs, and ResNets on NVIDIA RTX2080TIs (except **HET-XL** on ResNet, due to RAM usage).

First, we see that explicit uncertainty estimation does not come at a high RAM cost. The 3-layer MLP serving as uncertainty head for **MCInfoNCE**, **ELK**, **nivMF**, **HIB**, and **Losspred** adds 4.2M parameters to a ResNet or 2.6M to a ViT. The difference is due to the ViT's lower-dimensional embedding space. Bigger increases occur only for ensembles, which uses additional classifiers with $10 \cdot 2.1M$ parameters on ResNets and each $10 \cdot 0.5M$ on ViTs.

| | Model | Parameters (Millions) | Epoch Time (s) | Inference time per sample (ms) |
|---|---|---|---|---|
| **ResNet 50 on RTX 2080TI** | CE | 25.6 | 5971 | 3.7 |
| | InfoNCE | 23.5 | 6703 | 3.7 |
| | MCInfoNCE | 27.7 | 6656 | 3.8 |
| | ELK | 29.8 | 6703 | 3.8 |
| | nivMF | 29.8 | 6256 | 3.8 |
| | HIB | 29.8 | 7279 | 3.8 |
| | HET-XL (on A100) | 33.9 | (4189) | (2.3) |
| | Losspred | 29.8 | 6526 | 3.7 |
| | MCDropout | 25.6 | 7105 | 13.1 |
| | Ensemble | 46.0 | 6002 | 3.7 |
| | SNGP | 28.7 | 7632 | 3.7 |
| **ViT Medium on A100** | CE | 38.9 | 4922 | 2.8 |
| | InfoNCE | 38.3 | 7804 | 2.7 |
| | MCInfoNCE | 41.0 | 9883 | 2.4 |
| | ELK | 41.5 | 4838 | 2.7 |
| | nivMF | 41.5 | 4121 | 2.4 |
| | HIB | 41.5 | 3950 | 2.5 |
| | HET-XL | 39.4 | 4205 | 2.4 |
| | Losspred | 41.5 | 3814 | 2.6 |
| | MCDropout | 38.9 | 4721 | 9.6 |
| | Ensemble | 44.0 | 4107 | 2.5 |
| | SNGP | 42.0 | 4811 | 2.9 |

Table 1: Computational costs of all approaches. Epoch times include training on ImageNet as well as evaluating on all eight downstream datasets.

Training times should be interpreted with caution due to the aforementioned network storage. However, in general uncertainty estimates do not seem to exceedingly increase the train time, with between -22% and +28% over the CE baseline. Taking multiple samples during training (MCInfoNCE, nivMF, HIB, HET-XL) also does not systematically increase runtime. This is likely due to their efficient sampling implementations (Kirchhof et al., 2022; Davidson et al., 2018; Ulrich, 1984). The only consistent runtime cost occurs when training unsupervised models (InfoNCE, MCInfoNCE), which need to forward propagate two augmentations of each sample to obtain have positive pairs.

Similarly, providing an uncertainty estimate at inference takes only up to $0.1$ additional milliseconds on ResNets, because the uncertainties are all calculated within a single forward pass, and sampling was only required for the losses during training. The only exception here is MCDropout, which requires making 10 full forward passes during inference, which increases the time by a factor of roughly 3.6. The factor is not 10, because loading the data and storing the results is part of the measured elapsed time.

In summary, we find that uncertainty estimates only have small computational costs if they are implemented in a forward fashion.

