# OpenReview forum: "URL: A Representation Learning Benchmark for Transferable Uncertainty Estimates"
_NeurIPS.cc/2023/Track/Datasets_and_Benchmarks — NeurIPS 2023 Datasets and Benchmarks Poster_

### Official Review · Reviewer_JYVQ · 2023-07-05
**Uncertainty-aware representation learning is still an unsolved challenge**

**Rating:** 5
**Confidence:** 4
**Correctness:** The contribution claim in the paper i…
**Clarity:** The paper is well wrriten and easy to…

**Strengths:**

(1) The paper tackled a vital and challenge problem, URL.
(2) The proposed benchmarking method can help  to understand some uncertainty in representarion learning and measure it using the metric, R-AUROC；
(3）The authors conducted some experiments to demonstrate how to benchmark the uncertainty estimator for representation learning.


**Additional Feedback:**

N/A

**Documentation:**

The authors uploaded a package containing the source codes and dataset.  The package is download from their GitHub server. They are not available for public access so far.

**Ethics:**

No ethics issue.

**Limitations:**

The paper conducted some experiments on the same ground. TheThey did not touch the scaled data.

**Opportunities For Improvement:**

I donnot think there is a quick way to improve the quality under current research setting.


**Relation To Prior Work:**

The related work is well discussed.

**Summary And Contributions:**

Uncertainty quantification method is highly demanded in representation learning, either for transfer learning or  pretrained modes. This paper tried to develop a benchmarking method for the uncertainty-aware representation learning. With the proposed benchmarking metrics, R-AUROC, to evaluate the different uncertainty quantifiers. However, the URL still reminds an unsolved challenge.

---

> ### Author Response · Authors · 2023-08-18
>
> Dear reviewer,
>
> we would like to thank you for your time and feedback. Based on your and the other reviewers' comments we've added three ablation experiments, more discussions, and clarifying explanations to the manuscript. Let us know if there is any point in your or the other reviews you'd like us to comment on. We would be happy if our added experiments and discussions change your evaluation of our work to allow it to be accepted and accessible to the broad NeurIPS community.

---

### Official Review · Reviewer_qnX2 · 2023-07-12
**An extensive benchmark, but there may be problems with the overall setting**

**Rating:** 6
**Confidence:** 3

**Strengths:**

The benchmark presented in the paper is very extensive. The authors provide a detailed analysis of the results. A diverse array of uncertainty estimation methods is evaluated.

**Additional Feedback:**

A very minor error I found in the README.md of the supplementary material: The example class names for cub200 and cars196 in lines 50 and 56 are switched.

**Clarity:**

The paper is very well-written and easy to understand. The included figures enhance the text of the paper.

**Correctness:**

Apart from the general problems I see in the problem setting, the benchmark is conducted in an appropriate way.

**Documentation:**

The benchmark is documented very well. The supplementary material contains detailed information about the evaluated methods. Together with the code in the supplementary material, it should be possible to reproduce the results of the paper.

**Ethics:**

I did not find any issues that would require an ethics review.

**Limitations:**

The authors discuss some limitations of their benchmark in the paper, e.g. with respect to the used datasets and hyperparameter tunings in the experiments. The weaknesses/limitations I see (cf. "Opportunities for Improvement") are not discussed.

**Opportunities For Improvement:**

I see a problem with the general problem setting of the benchmark. The base assumption seems to be that uncertainties that transfer well should indicate whether predictions in a downstream task will fail. As the downstream datasets are by construction out-of-distribution with respect to the upstream dataset, I would expect that a good uncertainty predictor should predict high uncertainties across all samples of the downstream datasets. (In the literature on uncertainty estimation for classification, the capability to detect out-of-distribution samples is often used to judge the quality of uncertainty estimates.) It is unclear to me why one would expect a good uncertainty predictor to differentiate between different OOD samples with respect to whether their embeddings are close to OOD samples of the same class.

In addition, it is not clear to me whether it is useful to assign an uncertainty value to an embedding independent of a specific task. Whether an uncertainty estimate is good or not depends on the choice of the task. The embeddings might work very well for a certain task while being unsuitable for a different task on the same dataset. The uncertainties predicted for each sample would be the same, however.

A more specific problem I see about the chosen R-AUROC metric is that the "ground truth" positive and negative labels used to compute the false and true positive rate depend on the model embeddings. For example, one method might embed the downstream samples in a way that there are mostly positives (i.e., the nearest neighbor is of the same class), while another model might produce barely any positives. This makes the R-AUROC values of different models less comparable. A different choice of metric could alleviate this problem. (Finding such a metric is probably not a trivial task, however.)

**Relation To Prior Work:**

Related work is sufficiently discussed. The new contributions of the paper are explained clearly.

**Summary And Contributions:**

The authors introduce a benchmark for representation learning that evaluates the transferability of uncertainty estimates to zero-shot downstream tasks.
The used setting of representation and zero-shot learning is as follows:
- Models are trained on certain upstream tasks, e.g., classification on the ImageNet dataset or unsupervised settings.
- For the downstream evaluation, the authors test whether the learned embeddings transfer well to several zero-shot tasks. The used downstream datasets have classification labels that are used for the evaluation. The metric used for evaluating the transferability is the Recall@1 (R@1). For each embedding of a dataset sample, this metric computes whether the closest other embedding is of the same class. The Recall@1 is the rate of the nearest neighbor being of the same class across the whole dataset.

The methods evaluated produce uncertainty estimates on the upstream tasks computed in various ways. The uncertainty estimates a model predicts on downstream samples are used to compute a receiver-operating characteristic as follows:
- Based on the assumption that a high uncertainty should indicate that the R@1 is incorrect for a sample (i.e., the nearest neighbor embedding is of a different class), a threshold applied on the uncertainty produces a prediction whether the R@1 is correct (positive) or not (negative).
- By varying the uncertainty threshold, the resulting false positive and true positive rates form a ROC curve. The area under this ROC curve (named R-AUROC) is used to judge whether the uncertainties transfer well to the downstream task.

The authors provide an extensive benchmark of various kinds of models. The models are pretrained on the ImageNet dataset. Eight datasets are used for the downstream evaluation. The results show that no model achieves a good R-AUROC score (all below 0.6), indicating that there is much room for improvement.

The main contributions of the paper are
- the introduction of the problem of transferability of uncertainty estimates
- the definition of the R-AUROC metric to measure that transferability
- the conducted benchmark and its analysis

---

> ### Author Response · Authors · 2023-08-18
>
> Dear reviewer,
>
> we would like to thank you for your time and feedback. **TL;DR:** Based on your and the other reviewers' comments we've added three ablation experiments, more discussions, and clarifying explanations to the manuscript. We would be happy if this changed your evaluation of our paper to allow it to be accepted and accessible to the broad NeurIPS community.
>
> **Why would one expect a good uncertainty predictor to differentiate between different OOD samples? I would expect that a good uncertainty predictor should predict high uncertainties across all samples of the downstream datasets.**
>
> Thank you for this question. You are correct that predictive uncertainties are influenced by OODness, which is known as epistemic uncertainty. However, predictive uncertainty is also caused by aleatoric uncertainty, i.e., ambiguities in the inputs such as blurriness or low resolution. We expect a good pretrained model to detect these aleatoric uncertainties even within OOD samples, thereby differentiating them. We go into more details on this interesting question in Appendix A.3. Following your comment, we've also added ablations in Appendices C.4 and C.5 that show that the uncertainty estimates are indeed generally high on OOD data, as expected.
>
> **Is it useful to assign an uncertainty value to an embedding independent of a specific task? Whether an uncertainty estimate is good or not depends on the choice of the task.**
>
> Just as pretrained representations are good starting points for unknown downstream tasks, we seek uncertainty estimates that are good starting points for downstream uncertainty estimation. For this, the pretrained uncertainty estimators need to capture task-independent uncertainties, like blurriness or low-resolution of an input, which result in wrong predictions in any downstream task. In Figure 5, we show that our R-AUROC benchmark indeed correlates strongly (Rank Corr = 0.8) with how well a model captures those general notions of uncertainty, as well as with how uncertain human annotators are about images. We find your remark most intriguing and will add it to the FAQ section in Appendix A in the camera-ready version.
>
> **One method might embed the downstream samples in a way that there are mostly positives, while another model might produce barely any positives. This makes the R-AUROC values of different models less comparable.**
>
> This is not a problem when using the R-AUROC: It is correct that the R-AUROC depends on the individual errors of a model. This is intended, since a model's uncertainty is supposed to represent which of its own predictions are correct and which are wrong. However, the R-AUROC is independent of how many samples are correct or incorrect. A lower-accuracy model will not make it easier to achieve a high R-AUROC (e.g., by always predicting high uncertainties), because the R-AUROC examines the correct ranking within the uncertainty estimates as opposed to the absolute value. This is what makes it comparable between models and why it is widely used in uncertainty estimation literature. We comment on this in Appendices A.1 and A.2 and are happy to highlight this point in more detail in the camera-ready version.
>
> **Thanks again for reading and for your feedback!** Let us know if you have any more thoughts and if our added experiments and discussions change your evaluation of our work.

---

> > ### Comment · Reviewer_qnX2 · 2023-08-23
> > **Response to the authors**
> >
> > > **Why would one expect a good uncertainty predictor to differentiate between different OOD samples? I would expect that a good uncertainty predictor should predict high uncertainties across all samples of the downstream datasets.**
> > > \
> > > Thank you for this question. You are correct that predictive uncertainties are influenced by OODness, which is known as epistemic uncertainty. However, predictive uncertainty is also caused by aleatoric uncertainty, i.e., ambiguities in the inputs such as blurriness or low resolution. We expect a good pretrained model to detect these aleatoric uncertainties even within OOD samples, thereby differentiating them. We go into more details on this interesting question in Appendix A.3. Following your comment, we've also added ablations in Appendices C.4 and C.5 that show that the uncertainty estimates are indeed generally high on OOD data, as expected.
> > >  \
> > > **Is it useful to assign an uncertainty value to an embedding independent of a specific task? Whether an uncertainty estimate is good or not depends on the choice of the task.**
> > >  \
> > > Just as pretrained representations are good starting points for unknown downstream tasks, we seek uncertainty estimates that are good starting points for downstream uncertainty estimation. For this, the pretrained uncertainty estimators need to capture task-independent uncertainties, like blurriness or low-resolution of an input, which result in wrong predictions in any downstream task. In Figure 5, we show that our R-AUROC benchmark indeed correlates strongly (Rank Corr = 0.8) with how well a model captures those general notions of uncertainty, as well as with how uncertain human annotators are about images. We find your remark most intriguing and will add it to the FAQ section in Appendix A in the camera-ready version.
> >
> > Thank you for the further explanation and adding the ablation studies.
> >
> > Concerning aleatoric uncertainty, you mention blurriness and low resolution as examples for possible causes. I would argue that this is still task-dependent. It is easy to think of tasks for which blurriness and low resolution would not be a relevant issue. It might be fairer to say that one expects transferable uncertainty estimates to detect the kinds of causes for aleatoric uncertainty that were present in the upstream task also in downstream tasks.
> >
> > While the uncertainty estimates could possibly capture aleatoric uncertainty on downstream tasks (depending on the task in question, see above), I think that they will not be generally useful to capture epistemic uncertainty. Samples that are out-of-distribution with respect to the downstream task are not necessarily "more" out-of-distribution with respect to the original upstream task. Consequently, one cannot expect that the uncertainty estimates are higher for the OOD samples in that case (a trivial example is if we consider the upstream data as OOD with respect to the downstream data). In a sense, the epistemic uncertainty stays "anchored" to the original upstream task.
> >
> >
> > > **One method might embed the downstream samples in a way that there are mostly positives, while another model might produce barely any positives. This makes the R-AUROC values of different models less comparable.**
> > >  \
> > > This is not a problem when using the R-AUROC: It is correct that the R-AUROC depends on the individual errors of a model. This is intended, since a model's uncertainty is supposed to represent which of its own predictions are correct and which are wrong. However, the R-AUROC is independent of how many samples are correct or incorrect. A lower-accuracy model will not make it easier to achieve a high R-AUROC (e.g., by always predicting high uncertainties), because the R-AUROC examines the correct ranking within the uncertainty estimates as opposed to the absolute value. This is what makes it comparable between models and why it is widely used in uncertainty estimation literature. We comment on this in Appendices A.1 and A.2 and are happy to highlight this point in more detail in the camera-ready version.
> >
> > I agree with you that a lower-accuracy model does not have an inherent advantage. I intended to illustrate that, as you wrote, "the R-AUROC depends on the individual errors of a model". Therefore, the basis on which the metric is computed still varies from model to model. However, I recognize that the setting makes it hard (or even impossible?) to come up with a metric that avoids this problem without introducing new issues.

---

> > > ### Author Response · Authors · 2023-08-23
> > >
> > > Thank you for your thoughts! Let us briefly discuss your key points:
> > >
> > > **Aleatoric uncertainty, like blurriness and low resolution, is still task-dependent.**
> > >
> > > We agree that there could still exist tasks which require different uncertainty features than the pretrained ones. However, we think that learning uncertainties applicable to _most_ downstream tasks is still a valid quest, just like finding generally useful pretrained representations in traditional representation learning. The uncertainty features that are learned, like blurriness or low quality, of course have to be finetuned to the downstream task at hand, but this finetuning starts from a better starting point than if it has to be learned from scratch. The zero-shot performance of the pretrained uncertainties in Figure 1 is clearly above the baselines after all. In other words, for the many tasks where the community uses pretrained representations, pretrained uncertainties over those representations should also be applicable.
> > >
> > > **Epistemic uncertainty stays "anchored" to the original upstream task.**
> > >
> > > We fully agree. This is why our benchmark focusses on the aleatoric uncertainty. We've added the OOD experiments that were requested by the reviewers only as additional studies. In fact, their results support that some of the best suited pretrained uncertainty estimates indeed only estimate aleatoric and not epistemic uncertainty. (Appendix C.4, lines 209-212 in the revised version).
> > >
> > > **Is it possible to come up with uncertainty metrics independent of the model?**
> > >
> > > We believe it is in fact impossible to come up with uncertainty metrics independent of the individual errors of a model. This is because uncertainties are risk estimates, and risk is a function of the model's decisions. Hence, uncertainty estimates necessarily are dependent on the individual model's errors -- in fact all measures in the literature like ECE, log likelihood, Brier Score, and AUROC are. To present your important precautions better to the community, we will emphasize that the uncertainty performance our R-AUROC measures should always be analyzed along with the model performance, as we do in section 4.3.
> > >
> > > Please let us know if these updates change your evaluation of our work or if you see any other aspects to discuss.

---

> > > > ### Comment · Reviewer_qnX2 · 2023-08-24
> > > >
> > > > Thank you for the quick reply.
> > > >
> > > > > However, we think that learning uncertainties applicable to most downstream tasks is still a valid quest, just like finding generally useful pretrained representations in traditional representation learning. The uncertainty features that are learned, like blurriness or low quality, of course have to be finetuned to the downstream task at hand, but this finetuning starts from a better starting point than if it has to be learned from scratch.
> > > >
> > > > I understand your point. If the uncertainties transfer to a significant amount of "typical" downstream tasks, that is a valueable advantage. Do you have any insights into whether (few-shot)-finetuning the uncertainties on the downstream tasks actually yields good results? (This is not a request to perform any additional experiments in that direction for this paper.)
> > > >
> > > > > We believe it is in fact impossible to come up with uncertainty metrics independent of the individual errors of a model. This is because uncertainties are risk estimates, and risk is a function of the model's decisions. Hence, uncertainty estimates necessarily are dependent on the individual model's errors -- in fact all measures in the literature like ECE, log likelihood, Brier Score, and AUROC are.
> > > >
> > > > I think I agree with your point for metrics that measure aleatoric uncertainty. When I made my point, I was thinking of the standard OOD AUROC, which is based on the ground truth OOD labels (independent of the model) and the uncertainty scores (dependent on the model). This has makes comparison of models easier than the R-AUROC, for which the R@1 "labels" also depend on the model. However, the OOD AUROC measures measures epistemic uncertainty, thus avoiding the dependence on the model's errors.
> > > >
> > > > > To present your important precautions better to the community, we will emphasize that the uncertainty performance our R-AUROC measures should always be analyzed along with the model performance, as we do in section 4.3.
> > > >
> > > > I had some further thoughts regarding the question of whether a high-, medium-, or low-accuracy model has an advantage for R-AUROC. On average, this is not the case. However, a model with very high or very low accuracy (or R@1) has a greater chance at achieving unusually large or low R-AUROC values. If, for example, the uncertainty estimates are just random guesses (uniformly distributed between 0.0 and 1.0), the variance of resulting AUROC values is much larger for a model with accuracy 0.99 than for a model with accuracy 0.5. In both cases, the average R-AUROC is 0.5. At least in theory, this is a problem of the R-AUROC. The question is whether this is a significant problem in practice. Without having done a full analysis, I would suspect that at least some of the results in Figure 1 can be achieved with random uncertainty scores with a significant probability if the model is very accurate. However, its accuracy/R@1 would need to be much higher than that of all models you evaluated, so this might not be so relevant in this case. Nevertheless, I think it is good to keep these issues in mind, especially when comparing models with very different accuracies/R@1.
> > > >
> > > > Thank you for the interesting discussion and addressing my points. I am inclined to raise my rating.

---

> > > > > ### Author Response · Authors · 2023-08-25
> > > > >
> > > > > Thank you for the very interesting discussion, which we thoroughly enjoy!
> > > > >
> > > > > **Do you have any insights into (few-shot)-finetuning the uncertainties?**
> > > > >
> > > > > There is no literature on this, because pretraining and finetuning uncertainties is a novel undertaking. But reviewer HXyY has asked for few-shot results, so we've added some to Appendix C.6 in the revised appendix. We indeed see that finetuning is possible. As an example, the ELK loss achieves an R-AUROC of 0.585 at just 1 sample and monotonically increases with the number of samples. There are some counterexamples though: MCInfoNCE does not increase much over its zero-shot performance. We think this is because as an unsupervised loss it 1) has too few training samples in the few-shot setting and 2) does not learn the task-specific uncertainties. However, these results are of course preliminary and we plan further studies on this exact topic:
> > > > >
> > > > > Down the road, our goal is providing uncertainty estimates that are just trained along without requiring much attention from the downstream user. In this scope, the URL benchmark is the first step - finding good pretrained models - and the second step is indeed finetuning them. We are currently looking into the best practices for finetuning uncertainties in our follow-up work. We are excited to share more results in the near future!
> > > > >
> > > > > **Is the variance of the R-AUROC values for random guesses larger for a model with accuracy 0.99 than for a model with accuracy 0.5?**
> > > > >
> > > > > That's an interesting thought. We've ran your thought experiment in a quick simulation in R (code below). We find that the variance does not depend on R@1/accuracy of the model (see https://imgur.com/5EuKAAD). Overall, only 4 of the 1000 simulations gave R-AUROCs above 0.515. We think that these outliers can be caught by reporting variances, or, as we do in the paper, minima and maxima across several seeds.
> > > > >
> > > > > Let us know if you have further thoughts and thank you for the nice discussion. We're happy about your positive inclination!
> > > > >
> > > > >
> > > > > ```
> > > > > library(pROC)
> > > > >
> > > > > set.seed(0)
> > > > > p = seq(0.01, 0.99, 0.01)
> > > > > n_models_per_p = 10
> > > > > n_samples_per_model = 74557  # size of CUB + CARS + SOP test sets
> > > > >
> > > > > aurocs = matrix(0, nrow=length(p), ncol=n_models_per_p)
> > > > > for(p_idx in seq(1, length(p))){
> > > > >   for(t_idx in seq(1, n_models_per_p)){
> > > > >     # Simulate results for a model with p% accuracy / recall
> > > > >     is_correct = rbinom(n_samples_per_model, size=1, p=p[p_idx])
> > > > >
> > > > >     # Generate random uncertainties
> > > > >     uncertainty = runif(n_samples_per_model, min=0, max=1)
> > > > >
> > > > >     # Calculate AUROC
> > > > >     aurocs[p_idx, t_idx] = auc(is_correct, uncertainty)
> > > > >   }
> > > > > }
> > > > >
> > > > > png("RAUROC_random_baseline.png", width=640, height=480)
> > > > > plot(x = matrix(rep(p, n_models_per_p), nrow=n_models_per_p, byrow=TRUE),
> > > > >      y = aurocs, xlab = "R@1 of model", ylab = "R-AUROC of random uncertainty esimates")
> > > > > dev.off()
> > > > > ```

---

> > > > > > ### Comment · Reviewer_qnX2 · 2023-08-28
> > > > > >
> > > > > > Thanks a lot for pointing me to the few-shot results.
> > > > > >
> > > > > > Thanks for providing the simulation in R. I did a similar simulation in Python with a more finegrained list of target accuracies (from 0.001 tom 0.999). In the border cases (near 0.0 and 1.0), the R-AUROC with random guessing actually reached values far from 0.05 (above 0.7 and below 0.3). However, these are very rare events and probably will not matter much for real experiments. I do not think it is worth discussing this issue here in more detail.

---

### Official Review · Reviewer_D238 · 2023-07-21
**Review of Representation Learning Benchmark for Transferable Uncertainty Estimates**

**Rating:** 5
**Confidence:** 3
**Correctness:** Yes.

**Strengths:**

The paper makes a contribution to the advancement of pretrained models that offer reliable uncertainty estimates, a complex yet crucial problem in the machine learning domain. It creates a pathway for a more nuanced understanding of how models can predict uncertainties, which is a key step towards more reliable AI.

The thoroughness of this study is exemplary, with an evaluation of ten distinct uncertainty quantifiers that have been pre-trained on ImageNet and transferred to eight different downstream datasets. This comprehensive analysis provides an extensive review of the existing state-of-the-art methodologies. This breadth of experimentation enriches the study and deepens our understanding of uncertainty quantification, thus marking a substantial stride in this field of research.

**Additional Feedback:**

No.

**Clarity:**

This paper is moderately well wriiten. However, the excessive use of color fonts tends to cause a bit of distraction.

**Documentation:**

The level of detail provided in the paper is adequate to facilitate reproducibility.

**Ethics:**

No concern

**Limitations:**

The authors mentioned their limitations clearly.

**Opportunities For Improvement:**

While the evaluation of uncertainty quantifiers transferred to eight downstream datasets brings value, the results might not be universally applicable across different contexts. Broadening the variety of scenarios could enhance the robustness of the findings and their generalizability to a wider range of situations.

The paper's focus is mainly on classification tasks, leaving an ambiguity about how these evaluated uncertainty quantifiers might perform across other task types. Future research could benefit from considering an array of task types beyond classification such as segmentation, detection, OOD, to verify the adaptability of the quantifiers.

The study seems to overlook an analysis of the computational cost associated with the evaluated methods. A more holistic assessment should ideally incorporate such computational considerations to ensure the proposed approaches are both effective and efficient. This would enable practitioners to better gauge the feasibility of employing these methods in real-world applications.

**Relation To Prior Work:**

The authors assert that their research distinguishes itself from prior work in multiple aspects. These include the incorporation of pretrained models, the extensive evaluation of a multitude of uncertainty quantifications, and the novel approach of transferring these uncertainty estimates to subsequent downstream tasks. This unique blend of methodologies marks a departure from previous studies, thereby enriching the current discourse in this field.

**Summary And Contributions:**

Authors introduce an innovative benchmark named Uncertainty-aware Representation Learning (URL). This benchmark is designed to steer the evolution of pretrained models that deliver not only embeddings but also transferable uncertainty estimates. This study undertakes an evaluation of ten uncertainty quantifiers, all of which are pretrained on ImageNet and subsequently transferred to eight distinct downstream datasets. The empirical findings underscore the persistent challenge of achieving reliable uncertainty estimation, highlighting that both unsupervised and supervised methodologies have the capacity to learn transferable uncertainty estimates. Additionally, the authors also identify an interesting revelation; the quest for robust uncertainty estimation is not necessarily at odds with the objectives of conventional representation learning. This introduces a novel perspective in the existing body of research.

---

> ### Author Response · Authors · 2023-08-18
>
> Dear reviewer,
>
> we would like to thank you for your time and feedback. **TL;DR:** Based on your and the other reviewers' comments we've added three ablation experiments, more discussions, and clarifying explanations to the manuscript. We would be happy if this changed your evaluation of our paper to allow it to be accepted and accessible to the broad NeurIPS community.
>
> **Broadening the variety of scenarios to a wider range of situations such as OOD.**
>
> Thank you for the suggestion! We added an OOD scenario and a novel mixed ID+OOD uncertainty estimation scenario in Appendices C.4 and C.5. In the OOD scenario, the uncertainty estimates are tasked to provide higher uncertainties on the downstream CUB/CARS/SOP than on the upstream ImageNet data.
> In the mixed scenario, models have to provide uncertainty estimates on a mixture of ImageNet and CUB/CARS/SOP data. This tasks both their ability to assign higher uncertainties to OOD samples and to correctly rank their uncertainties within the ID and within the OOD data.
> We find that both ResNets and ViTs generally put OOD embeddings closer to the origin of the embedding space. Thus, uncertainty estimators that use the norm of the embedding vector for their uncertainty estimates either directly (InfoNCE) or indirectly (cross-entropy-based methods where a small norm leads to more uniform class probabilities) perform the best in both scenarios, notably SNGP and HET-XL.
>
> **An analysis of the computational cost associated with the evaluated methods.**
>
> Thank you for bringing up this important point. We have added the number of parameters, train times, and inference times per sample to Appendix C.7. In summary, we find that the number of parameters is increased by 0-33\% compared to a cross-entropy baseline, except for ensembles, which double the parameters. However, train times per epoch are largely unaffected by this (-22\% to +28\%) even when models take samples for their uncertainties. Since most models obtain their uncertainties via simple forward passes during inference, it increases their inference time merely from 3.7ms to 3.8ms on ResNets (except for MCDropout, where the sampling at inference time leads to a 3.6x increase). We will highlight these points in the camera-ready version.
>
> **Thanks again for reading and for your feedback!** Let us know if you have any more thoughts and if our added experiments and discussions change your evaluation of our work.

---

### Official Review · Reviewer_HXyY · 2023-07-22

**Rating:** 6
**Confidence:** 3
**Clarity:** This paper is well written with solid…

**Strengths:**

(1) **Contribution:** This paper introduces a novel URL benchmark to evaluate state-of-the-art uncertainty estimators on unseen data. The observations from the experimental results reveal the connections between uncertainty estimation and embedding estimation. It also points out that models with good uncertainties upstream might not be a solid indicator of downstream.

(2) **Relevance:** The proposed benchmark demonstrates that transferable uncertainty estimation is challenging and unsolved for existing approaches. Thus, it provides some insights into generalizing the estimated uncertainty to unseen data. This might motivate the community to design universal uncertainty estimators in future work.

(3) **Quality:** This paper is well written and the problem setting of transferable uncertainty estimation is well motivated. The main findings are supported by the experimental results.

(4) **Ethical and social implications:** There is no explicit ethical concern for the proposed benchmark. Instead, transferable uncertainty estimation allows for exploring the trustworthiness of the model in unseen data and domains.

**Additional Feedback:**

(1) More insights can be provided for the observations in the experiments. It shows that uncertainty estimation is not always in conflict with embedding estimation. The correlation between uncertainty estimation and embedding estimation can be further explained.

(2) The transferability problem URL can be related to task transferability (e.g., task similarity between upstream and downstream tasks) and the number of training samples in downstream tasks (e.g., a large number of samples might provide a better uncertainty estimator). This can further be discussed in the experiments.

**Correctness:**

This paper developed a novel URL benchmark. The evaluation methods and experiment design of this paper were appropriate and performed correctly.

**Documentation:**

This paper provided sufficient detail to support reproducibility.

**Limitations:**

The authors adequately addressed the limitations and potential negative societal impact of their work.

**Opportunities For Improvement:**

(1) Some related works [r1] on transferable uncertainty estimates are missing. The major contributions of this paper should be emphasized compared to these related works.

[r1] Ovadia, Yaniv, Emily Fertig, Jie Ren, Zachary Nado, David Sculley, Sebastian Nowozin, Joshua Dillon, Balaji Lakshminarayanan, and Jasper Snoek. "Can you trust your model's uncertainty? Evaluating predictive uncertainty under dataset shift." Advances in neural information processing systems 32 (2019).

(2) The correlation between the performance of transferable uncertainty estimation and task transferability. For example, it is unclear whether increasing task transfer difficulty would also reduce the quality of the uncertainty estimator.

(3) The upper reference in section 4.2 can be further discussed by considering different numbers of training samples. When limited training samples are given, existing methods might show comparable performance to the upper reference.



**Relation To Prior Work:**

Some related works on transferable uncertainty estimation are missing. The contributions of this paper can further be explained compared to these works.

[r1] Ovadia, Yaniv, Emily Fertig, Jie Ren, Zachary Nado, David Sculley, Sebastian Nowozin, Joshua Dillon, Balaji Lakshminarayanan, and Jasper Snoek. "Can you trust your model's uncertainty? Evaluating predictive uncertainty under dataset shift." Advances in neural information processing systems 32 (2019).

**Summary And Contributions:**

This paper proposed a new Uncertainty-aware Representation Learning (URL) benchmark for transferable uncertainty estimates. It introduces an easy-to-implement metric to evaluate the quality of uncertainty quantification on unseen downstream data. The experimental results show that transferable uncertainty estimation is challenging for existing in-distribution uncertainty quantification approaches. Furthermore, it is observed that URL captures how aligned a model is with human uncertainty.

---

> ### Author Response · Authors · 2023-08-18
>
> Dear reviewer,
>
> we would like to thank you for your time and feedback. **TL;DR:** Based on your and the other reviewers' comments we've added three ablation experiments, more discussions, and clarifying explanations to the manuscript. We would be happy if this changed your evaluation of our paper to allow it to be accepted and accessible to the broad NeurIPS community.
>
> **Related works on transferable uncertainty estimates (Ovadia et al.)**
>
> Thank you for providing the reference, which we will discuss in the related works section of the camera-ready version. The key difference is that Ovadia et al. see uncertainty estimation from a robustness perspective. They work towards calibrated uncertainty estimates on corrupted upstream (ID) data and want generally high uncertainties on OOD data for OOD detection. We take a transferability perspective. Pretrained models should not just predict high uncertainties on the OOD data they are deployed on, but order how uncertain they are between the samples. This is also why we apply different metrics (R-AUROC) and approaches (probabilistic embeddings, risk prediction) than Ovadia et al.. This being said, we have taken your comment as an inspiration to add OOD detection experiments in the style of Ovadia et al., along with novel experiments on ID+OOD data that blend their and our work in Appendices C.4 and C.5. We find that cross-entropy based approaches are good in assigning higher uncertainties to OOD data but cannot order within OOD data, whereas probabilistic embeddings provide better orders within ID and within OOD data but are not as good in assigning generally higher uncertainties to OOD data than to ID data.
>
> **Would increasing task transfer difficulty also reduce the quality of the uncertainty estimator?**
>
> This is an interesting question. We do not have a quantitative measure of task transfer hardness, but can compare the difficulty of benchmarks in literature. We see four categories: In-distribution (our upstream R-AUROC), Corruptions (Ovadia et al.'s benchmark), transfer to new classes on the same dataset (Deep Metric Learning benchmarks, e.g., A Non-isotropic Probabilistic Take on Proxy-based Deep Metric Learning, Kirchhof et al.), and transfer to new datasets (our downstream R-AUROC). Across these benchmarks (and within the ablations of each of them), we indeed see that the performance lowers with increasing shift. We hope that our benchmark adds to these efforts as the so far hardest transferring challenge.
>
> **The upper reference in section 4.2 can be further discussed by considering different numbers of training samples [...] in downstream tasks.**
>
> Thank you for the suggestion. We have added few-shot experiments for both the cross-entropy baseline and the pretrained uncertainty estimators. First, we find that the current zero-shot R-AUROCs match those of 2-5 shot cross-entropy baselines (equaling 200-500 total samples on CUB/CARS and 20000-50000 on SOP). Second, additional few-shot samples increase the performance of the pretrained uncertainty estimators, as expected. Feel free to find more details in Appendix C.6.
>
> **Thanks again for reading and for your feedback!** Let us know if you have any more thoughts and if our added experiments and discussions change your evaluation of our work.

---

### Author Response · Authors · 2023-08-28

Dear reviewers, dear area chair,

thank you for your time and the interesting discussion which we are happy to continue for the last two days of the discussion period. We are glad that you also find that a pretrained _"uncertainty quantification method is highly demanded in representation learning"_ (JYVQ) and that our work _"introduces a novel perspective in the existing body of research"_ (D238). We are happy that our benchmark _"tackled [the] vital and challenging problem"_ (JYVQ) _"to design universal uncertainty estimators"_ (HXyY) _"which is a key step towards more reliable AI"_ (D238). We are humbled to hear that _"the thoroughness of this study is exemplary"_ (D238) with a _"diverse array of uncertainty estimation methods"_ (qnX2). To add to the _"detailed analysis of the results"_ (qnX2), we gratefully enhanced the manuscript with your feedback. In summary, we

* Added the requested OOD, OOD+ID, few-shot, and runtime studies
* Benchmarked a further method and divergence-based uncertainties for ensembles
* Will add intuitions and answers to our discussion and FAQ section to the camera-ready version
* Will add the related work on OOD detection to the camera-ready version

We would be happy to hear if these additions properly address your thoughts and update your evaluation of our paper.

---

### Decision · Program_Chairs · 2023-09-22

**Decision:**

Accept (Poster)

**Comment:**

The authors introduce a benchmark for representation learning that evaluates the transferability of uncertainty estimates to zero-shot downstream tasks. The authors provide an extensive benchmark of various kinds of models. The models are pre-trained on the ImageNet dataset. Eight datasets are used for the downstream evaluation. The results show that no model achieves a good R-AUROC score (all below 0.6), indicating much room for improvement.

If looking at the average score (6,5,6,5), the mean 5.5. doesn't place this paper competitively above the acceptance bar. However, I am recommending acceptance based on the following facts:

- Reviewer JYVQ (rate 5) provided literally no helpful information in his/her review, and did not participate in any follow-up discussion with either author or AC.  AC firmly believes counting on such an outlier is damaging to a fair conference. So, the rating from Reviewer JYVQ was excluded, leaving scores (6,6,5) as "factual ratings" that AC considers.

- Reviewer D23 (another rate 5) did not participate in followup discussion. AC read his/her comments and found his main questions, namely the variety of scenarios including OOD and the computational cost issues, have been thoroughly addressed by the authors. The former issue was same raised by Reviewer qnX2, and was addressed to satisfaction by authors (as confirmed by Reviewer qnX2)

- Reviewers HXyY and qnX2 (both rate 6) provide longer and informative review comments. Particularly, Reviewer qnX2 engaged in thorough, multi-round discussions with the authors, and they came to many consensus afterward. Reviewer qnX2 actually improved the rating after the discussion to show support.


Comparing all papers in AC' batch, it is believed that this paper has the highest quality, and the authors addressed the reviewers' comments to the best. For example, the most engaging reviewer (qnX2) has raised many constructive concerns and the authors resolved them all. Therefore, AC found it fair to consider this work a competitive, and ready contribution to NeurIPS D&B.

This decision has been made in agreement with the AC and SAC.